# Robust bilayer solid electrolyte interphase for Zn electrode with high utilization and efficiency

Yahan Meng[1,4], Mingming Wang[1,4], Jiazhi Wang[2,4], Xuehai Huang[3], Xiang Zhou[3], Muhammad Sajid[1], Zehui Xie[1], Ruihao Luo[1], Zhengxin Zhu[1], Zuodong Zhang[1], Nawab Ali Khan[1], Yu Wang[3] ✉, Zhenyu Li[2] ✉ & Wei Chen[1] ✉

Construction of a solid electrolyte interphase (SEI) of zinc (Zn) electrode is an effective strategy to stabilize Zn electrode/electrolyte interface. However, single-layer SEIs of Zn electrodes undergo rupture and consequent failure during repeated Zn plating/stripping. Here, we propose the construction of a robust bilayer SEI that simultaneously achieves homogeneous $Zn^{2+}$ transport and durable mechanical stability for high Zn utilization rate (ZUR) and Coulombic efficiency (CE) of Zn electrode by adding 1,3-Dimethyl-2-imidazolidinone as a representative electrolyte additive. This bilayer SEI on Zn surface consists of a crystalline $ZnCO_3$-rich outer layer and an amorphous ZnS-rich inner layer. The ordered outer layer improves the mechanical stability during cycling, and the amorphous inner layer homogenizes $Zn^{2+}$ transport for homogeneous, dense Zn deposition. As a result, the bilayer SEI enables reversible Zn plating/stripping for 4800 cycles with an average CE of 99.95% (± 0.06%). Meanwhile, Zn||Zn symmetric cells show durable lifetime for over 550 h with a high ZUR of 98% under an areal capacity of 28.4 mAh $cm^{-2}$. Furthermore, the Zn full cells based on the bilayer SEI functionalized Zn negative electrodes coupled with different positive electrodes all exhibit stable cycling performance under high ZUR.

Lithium-ion batteries (LIBs) have been successfully and widely applied in electric vehicles and portable electronics due to their high energy density[1,2]. Nevertheless, poor safety of organic electrolyte and high cost of lithium (Li) restrict their further development into the field of large-scale energy storage[3,4]. Aqueous batteries, in particular aqueous zinc ion batteries (ZIBs), have been emerged as a promising alternative to LIBs due to their inherent safety, low cost and non-toxicity[5,6]. Metallic zinc (Zn) electrode has the advantages of high

volumetric capacity (5855 mAh $cm^{-3}$), mild redox potential (−0.763 vs standard hydrogen electrode), and good reversibility of Zn plating/stripping[7,8]. Unfortunately, Zn electrodes still suffer from uncontrollable dendrites formation and severe side reactions due to the unstable electrode/electrolyte interface, hindering the cycling life and reversibility of Zn plating/stripping (Fig. 1a)[9,10]. Therefore, it is critical to construct stable electrode/electrolyte interface for Zn electrode.

[1]Department of Applied Chemistry, School of Chemistry and Materials Science, Hefei National Research Center for Physical Sciences at the Microscale, University of Science and Technology of China, Hefei, Anhui, China. [2]Key Laboratory of Precision and Intelligent Chemistry, University of Science and Technology of China, Hefei, Anhui, China. [3]Center for Electron Microscopy, South China Advanced Institute for Soft Matter and Guangdong Provincial Key Laboratory of Functional and Intelligent Hybrid Materials and Devices, School of Emergent Soft Matter, South China University of Technology, Guangzhou, China. [4]These authors contributed equally: Yahan Meng, Mingming Wang, Jiazhi Wang. ✉e-mail: roywangyu@scut.edu.cn; zyli@ustc.edu.cn; weichen1@ustc.edu.cn

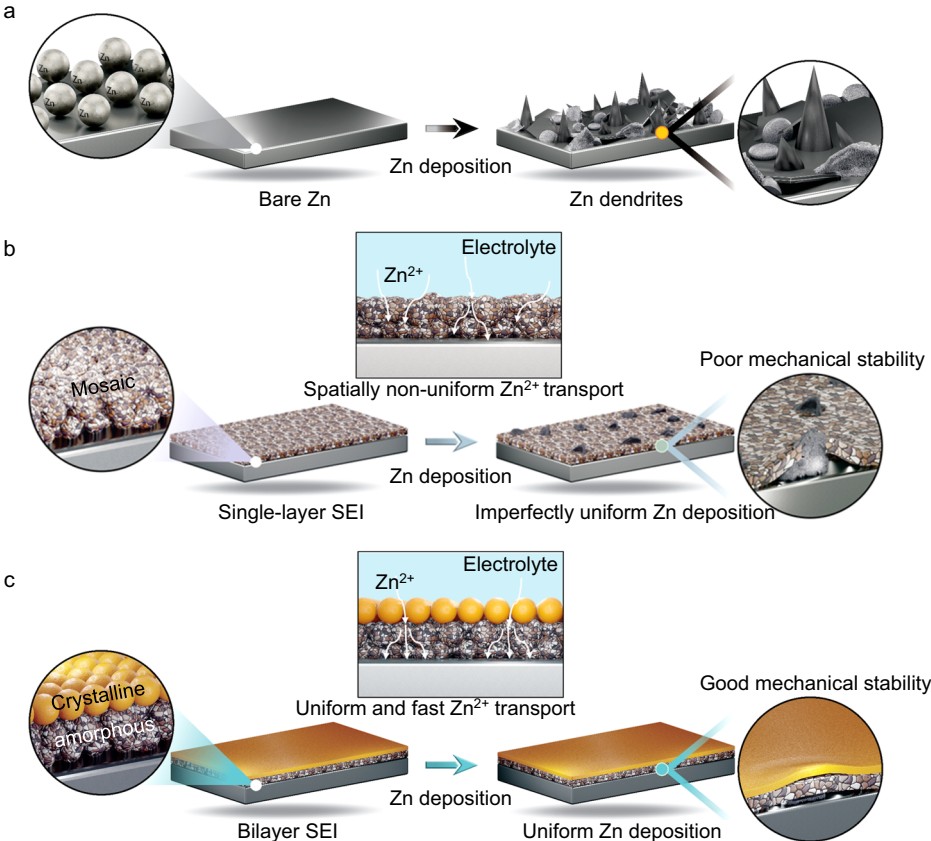

**Fig. 1 | Schematic diagram of the functions of different SEIs on Zn deposition. a** Structural evolution on bare Zn during deposition. **b** Structural evolution on single-layer SEI during deposition. **c** Structural evolution on bilayer SEI during deposition.

To date, different strategies such as electrode surface modification, solvation structure regulation, and electrolyte additives have been adopted to stabilize electrode/electrolyte interface for Zn[11,12]. Among them, the construction of a high-quality solid electrolyte interphase (SEI) through electrolyte component regulation is undoubtedly considered as a promising and effective strategy to stabilize Zn electrodes[13]. It has been shown that SEI of Zn electrode not only effectively reduces side reactions such as hydrogen evolution reaction (HER) and corrosion problems induced by direct contact between active water molecules and Zn surface, but also promotes uniform plating/stripping of Zn and greatly improves the Coulombic efficiency (CE) by homogenizing Zn[2+] flux[14,15]. Previous work has been devoted to the development of stable SEI layers on Zn electrodes. For instance, Wang et al.[16] reported a $ZnF_2$-rich SEI with trimethylethyl ammonium trifluoromethanesulfonate ($Me_3EtNOTF$) additive, which could achieve the CE of 99.9%. Nazar et al.[17] introduced N,N-dimethylformamidium trifluoromethanesulfonate (DOTf) in a zinc sulfate electrolyte, leading to the formation of a fluoride-free SEI, which realized highly stable and reversible Zn plating/stripping. Yang et al.[18] used hexamethylphosphoric triamide as an electrolyte additive to achieve a high Zn utilization rate (ZUR) of 85.6%. Despite of the remarkably improved reversibility and stability of Zn electrode owing to SEI, the structures of the previously reported SEI were dominant in single layers.

It is generally recognized that the single-layer mosaic structure of SEI is described as a non-homogeneous distribution of inorganic and organic components, i.e., it is mainly characterized by a random and uniform distribution of inorganic crystalline components within the overall amorphous structure[19]. Single-layer SEI of Zn electrode based on mosaic structures can obviously promote the lifetime and inhibit the growth of Zn dendrites compare to bare Zn electrode (Fig. 1a)[20].

However, the random distribution of inorganic components in the mosaic structure has spatially heterogenized Zn[2+] transport, leading to unfavorable and uneven Zn plating/stripping processes[21]. As a result, the uneven stress distribution of heterogeneous Zn plating/stripping is likely to cause the fracture of SEI during cycling (Fig. 1b). Therefore, advanced design of SEI of Zn electrode is still urgently needed for further in-depth study.

An "ideal" SEI should possess not only good Zn[2+] conductivity and electronic insulation but also a high mechanical modulus, remaining stable and unyielding under stress or volume changes[22]. It is well known that the homogeneity and mechanical stability of the SEI are equally critical for the metal plating/stripping[23]. Moreover, the structure of the SEI plays a decisive role in its homogeneity and mechanical stability[24]. Compared with the mosaic single-layer SEI, ordered bilayer SEI can minimize the mutual interference among different components, as can be learnt from Li SEI in Li-ion batteries[4]. We hypothesize that the ordered nature of the bilayer structure can provide a more homogeneous Zn[2+] transport, which can promote the uniform deposition of Zn, while maintaining a high mechanical stability, making it more durable and less prone to breakage during the battery cycling process (Fig. 1c).

In this work, we demonstrate a structure design of a robust amorphous/crystalline bilayer SEI that simultaneously achieves the desirable homogeneous Zn[2+] transport and mechanical stability for Zn electrode chemistry to achieve high ZUR and CE. Characterizations such as statistical transmission electron microscopy (TEM), X-ray photoelectron spectroscopy (XPS) and time-of-flight secondary ion mass spectrometry (ToF-SIMS) reveal that a bilayer SEI on Zn surface consisting of a crystalline $ZnCO_3$-rich outer layer and an amorphous ZnS-rich inner layer can be achieved by adding 1,3-Dimethyl-2-imidazolidinone (DMI) as an electrolyte additive. The structure of inorganic

crystalline outer layer and overall elastic C/N-rich organics improve the mechanical stability of SEI during cycling. Moreover, the amorphous inner layer homogenizes $Zn^{2+}$ transport, realizing homogeneous, dense Zn deposition and prominent reversibility. The synergistic bilayer SEI enables a high average CE (99.95%) of Zn plating/stripping for 4800 cycles at $1\,mAh\,cm^{-2}$ and $2\,mA\,cm^{-2}$. Meanwhile, Zn||Zn symmetric cells can achieve stable cycling for more than 550 h with a high ZUR of 98% under a large areal capacity of $28.4\,mAh\,cm^{-2}$ ($2\,mA\,cm^{-2}$). Moreover, the full cells based on the bilayer SEI functionalized Zn negative electrode all show improved cycling performance when paired with different positive electrodes under low negative to positive capacity (N/P) ratio. For example, the Zn||$I_2$ full cell maintains stable cycling for more than 5000 cycles with negligible capacity degradation (from $209\,mAh\,g^{-1}$ to $200.8\,mAh\,g^{-1}$ at $2.1\,A\,g^{-1}$) at a restricted N/P ratio of 1.5. The anode-free Zn||Br pouch cell displays a durable lifespan of 800 cycles without any pre-stored Zn.

## Results

DMI, an industrially used nonprotonic solvent with good solvency, low toxicity, low viscosity and high dielectric constant was used an electrolyte additive that partially replaces water molecules and thereby changes the solvation structure of $Zn^{2+}$[25]. In this work, we designed to use only trace amounts of DMI (10-100 mM) to form the SEI of Zn electrode without changing the solvation structure of the $Zn^{2+}$ ions. Furthermore, we calculated the energy levels of the lowest unoccupied molecular orbital (LUMO) and the highest occupied molecular orbital (HOMO) for DMI and $H_2O$. The results show that DMI has a narrower electrochemical stabilization window (ESW) compared to $H_2O$ (Supplementary Fig. 1)[14,26]. In particular, the LUMO energy level of DMI of 1.64 eV is lower than that of $H_2O$ (1.78 eV), suggesting that DMI is preferentially reduced and potentially generated SEI on Zn electrode during the electrochemical process. Considering all factors, an optimal concentration of DMI was determined to be 10 mM in the 2 M $ZnSO_4$ electrolyte (Supplementary Figs. 2-3). In addition, we characterized the different electrolytes (2 M $ZnSO_4$ and 2 M $ZnSO_4$ + 10 mM DMI) using Raman and Fourier transform infrared (FT-IR) in order to verify whether the solvated structures of electrolytes are changed. As shown in Supplementary Figs. 4-5, the Raman and FT-IR spectra of $ZnSO_4$-DMI and $ZnSO_4$ electrolytes exhibit negligible difference, indicating that a small amount of DMI as an additive does not affect the solvated structure of $Zn^{2+}$[9]. Furthermore, the viscosity and ionic conductivity of the $ZnSO_4$-DMI electrolyte change slightly compared to the pure $ZnSO_4$ electrolyte, which is an advantage of the trace amount of DMI as the additive (Supplementary Fig. 6).

### Nanostructure of bilayer SEI

The nanostructure of SEI was further determined via high-resolution transmission electron microscopy (HRTEM). The TEM sample preparation process is detailed in the Methods section and Supplementary Fig. 7. As shown in Fig. 2a and Supplementary Fig. 8, no interfacial layer can be found in Zn deposited in pristine $ZnSO_4$ electrolyte, because the decomposition products of solvated $H_2O$ cannot form a dense layer to cover the surface of Zn electrode[15,27,28]. However, as shown in Fig. 2b and Supplementary Fig. 9, Zn deposited in $ZnSO_4$-DMI electrolyte has a distinct interfacial layer with a homogeneous SEI about 25 nm in thickness. Further investigations reveal that the SEI consists of two distinct layers, where the outer layer is crystalline and the inner layer is amorphous (Fig. 2c). The crystalline outer layer can provide good mechanical stability and the amorphous inner layer can homogenize $Zn^{2+}$ transport through SEI, which enables a mechanically and electrochemically robust SEI for Zn electrode. Supplementary Figs. 10-11 exhibit the lattice fringes corresponding to $ZnCO_3$ (104) in the outermost crystalline layer, which correspond to Zn (002) in the deposited Zn bulk. The high-angle annular dark-field (HAADF) image (Fig. 2d) with the corresponding energy dispersive spectroscopy (EDS)

mapping reveal that the elements of C, S, N, O, and Zn are uniformly distributed over the entire SEI layer (Fig. 2e–i).

Due to the multifarious possibilities of SEI components and the complexity of HRTEM images, it is not reliable enough to ascribe a crystal structure from an image by simply measuring its lattice parameters and matching with the existing ones[23]. Thorough structural identification requires the consideration of all possible projections of reasonable structures and careful comparison of lattice information from images. Therefore, we selected three representative regions (referred to as Areas 1–3) from the HRTEM images and conducted a thorough structural comparison with 187 related crystal structures in the Materials Project and Inorganic Crystal Structure Database (ICSD). We performed a fast Fourier transform on each image (Fig. 2j–l) and quantitatively evaluated their similarity to the cross-sections of every crystal in reciprocal space. Statistical analyses show that the images of the crystalline region of SEI closely match with three structures among the studied 187 ones: ZnO-mp-2133, $ZnCO_3$-mp-9812, and $Zn(CO_2)_2$-mp-559437 (Fig. 2m–u and Supplementary Figs. 12-17). The calculated structure of $Zn(CO_2)_2$-mp-559437 is unlikely to form in our experiment due to the high formation energy. The presence of ZnO may result from the rapid oxidation of the sample exposure to air[29,30]. Therefore, based on the above characterizations, the crystalline outer layer of the bilayer SEI is defined to be $ZnCO_3$-rich.

### Chemical composition and mechanical stability of bilayer SEI

In order to clarify the composition of bilayer SEI, the depth-profiled composition was firstly investigated by X-ray photoelectron spectroscopy (XPS) coupled with $Ar^+$ sputtering from 0 s to 600 s. The surface of the bilayer SEI layer contains significant C 1s signals that can be deconvoluted into three species. The peak at 288.6 eV is attributed to the $ZnCO_3$, while the peaks at 285.4 and 284.7 eV are attributed to the C-O/C-N and C-C/C-H bonds, respectively[17]. We attribute these organic species to the electrochemical decomposition/surface adsorption of DMI. As the $Ar^+$ sputtering consistently proceeds, the signals of C-C/C-H and C-O/C-N species drop sharply, and the signals of $ZnCO_3$ species also become weaker in the spectra (Fig. 3a). This indicates that the outer layer of SEI is comprised of $ZnCO_3$ and C-rich organic components, which is consistent with the TEM analysis of crystalline outer layer. It can be seen from N 1s spectra that significant signals of pyrrolic C-N at 400.4 eV and pyridinic C-N at 398.8 eV appear before and after $Ar^+$ sputtering (Fig. 3b)[31]. Pyridinic C-N species are formed through the electrochemical decomposition of DMI and pyrrolic C-N species are associated with DMI adsorption[14]. It is observed that the ratios of pyridinic C-N/pyrrolic C-N increase as the sputtering process. This finding suggests that the large number of organic components inside the bilayer SEI is mainly due to the electrochemical decomposition of DMI rather than simple surface adsorption. Furthermore, the N content does not decrease but increases slightly upon sputtering to 600 s, indicating that a uniform and dense N-rich organic layer is formed throughout the overall SEI layer. This overall C/N-rich organic structure can be adaptable to large volume changes during Zn plating/ stripping due to high viscoelasticity[4]. As shown in Fig. 3c, the S 2p spectrum before $Ar^+$ sputtering confirms the presence of significant amounts of $SO_4^{2-}$ with weak ZnS signal, which represents the outer layer of SEI is essentially free of ZnS[32]. However, the amount of ZnS increases significantly with the sputtering depth, suggesting that the inner layer of SEI is ZnS-rich. ZnS and trace amounts of $SO_3^{2-}$ may be produced by the reduction of $SO_4^{2-}$[33]. In contrast, the Zn foil cycled in $ZnSO_4$ electrolyte shows almost no signals of C and S elements after sputtering, further confirming that no SEI is formed in $ZnSO_4$ electrolyte (Supplementary Fig. 18). Meanwhile, the inner layer of the bilayer is amorphous since ZnS, C/N rich organics, and $ZnCO_3$ together form a mixed phase. Furthermore, the percentage of C, N, and S elements with depth also verifies the difference between the inner and outer components of SEI (Supplementary Fig. 19). Based on these XPS

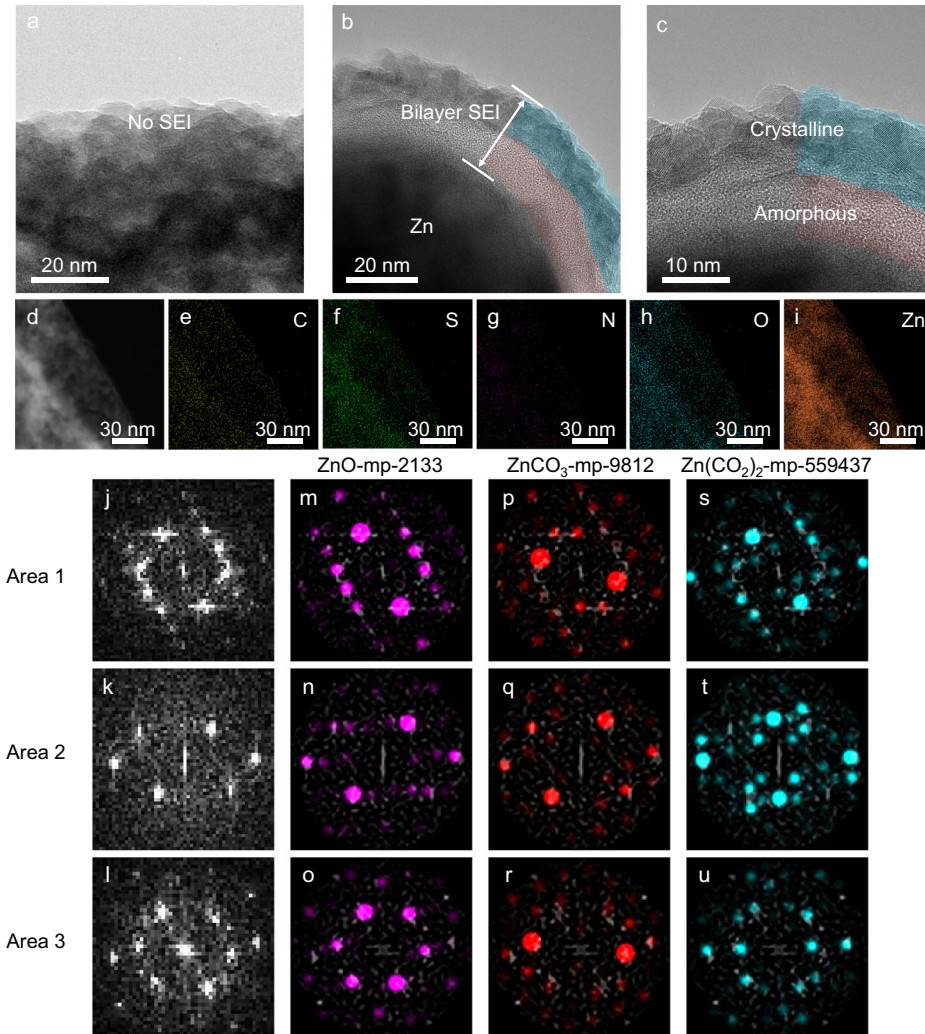

**Fig. 2 | Characterization of nanostructure and composition of SEI. a** HRTEM images of Zn surface with no SEI formation in ZnSO$_4$ electrolyte. **b**, **c** HRTEM images of bilayer SEI on the Zn surface in ZnSO$_4$-DMI electrolyte. **d** STEM image of bilayer SEI on the Zn surface. **e**–**i** The corresponding EDS mapping of (**d**). **j**–**l** Fast Fourier transform on three representative regions of the SEI outer layer. **m**-**u**, Actual vs. theoretical reciprocal space of (**m**–**o**) ZnO-mp-2133, (**p**–**r**) ZnCO$_3$-mp-9812 and (**s**–**u**) Zn(CO$_2$)$_2$-mp-559437 in different crystal regions.

results, we summarize that the bilayer SEI is composed of the crystalline ZnCO$_3$-rich outer layer and amorphous ZnS-rich inner layer, while its overall components are dominated by C/N-rich organics.

The three-dimensional (3D) spatial distribution of each component of the bilayer SEI was further investigated by the depth profile analysis of time-of-flight secondary ion mass spectrometry (ToF-SIMS). CH$^-$ and CN$^-$ are characteristic ionic fragments of organic components, while CO$_3^-$ and ZnS$^+$ are characteristic ionic fragments of ZnCO$_3$ and ZnS components, respectively. With the sputtering time increasing, the content of CH$^-$ drops rapidly and almost disappears, which is concordant with C 1 s spectrum. In addition, the CN$^-$ organic components in SEI distribute uniformly along the thickness and are likely compact to form a robust layer, which is in good buffering and connectivity for the entire SEI (Fig. 3d). It can be concluded that the signal of Zn sulfide (Zn$_x$S$_y$) shows a slightly increased trend and that of CO$_3^{2-}$ shows a downward trend with the sputtering time, which is in good agreement with the XPS results (Supplementary Fig. 20). Thus, combining the XPS, ToF-SIMS, and TEM results, we conclude that in the DMI additive electrolyte, Zn electrode can form a bilayer SEI structure with the ZnCO$_3$-rich outer layer, ZnS-rich inner layer and overall C/N-rich organic bulk phase. This

bilayer SEI with the balanced rigidity of inorganics and viscoelasticity of organics is able to maintain good homogeneity of Zn$^{2+}$ transport and mechanical stability during cycling.

Inorganic SEI with high mechanical modulus can counter against the stress from the dendrite growth. Mechanical stability of the bilayer SEI was quantified by the measurement of Derjaguin-Müller-Toporov (DMT) modulus by atomic force microscopy (AFM)[4,34]. The average DMT modulus of the bilayer SEI is as high as 61.2 GPa, which is ascribed to the outer layer of the bilayer SEI that is in rich of inorganic components (Fig. 3e, f)[35]. These AFM results reveal the mechanical stability of the bilayer SEI that can be well maintained during repeated Zn plating/stripping due to the synergy between organic and inorganic layers, leading to good durability against SEI rupture.

### Formation mechanism of bilayer SEI
Based on the above experimental results, it can be concluded that the bilayer SEI structure can be formed on the surface of the Zn electrode by adding DMI into the electrolyte. It consists of an outer crystalline layer and an inner amorphous layer, in which the outer layer is ZnCO$_3$-rich and the inner layer is ZnS-rich, and the whole SEI layer is of uniform C/N organics-rich (Fig. 4a). Inorganic compositions with high mechanical strength can inhibit dendrite formation, and organic

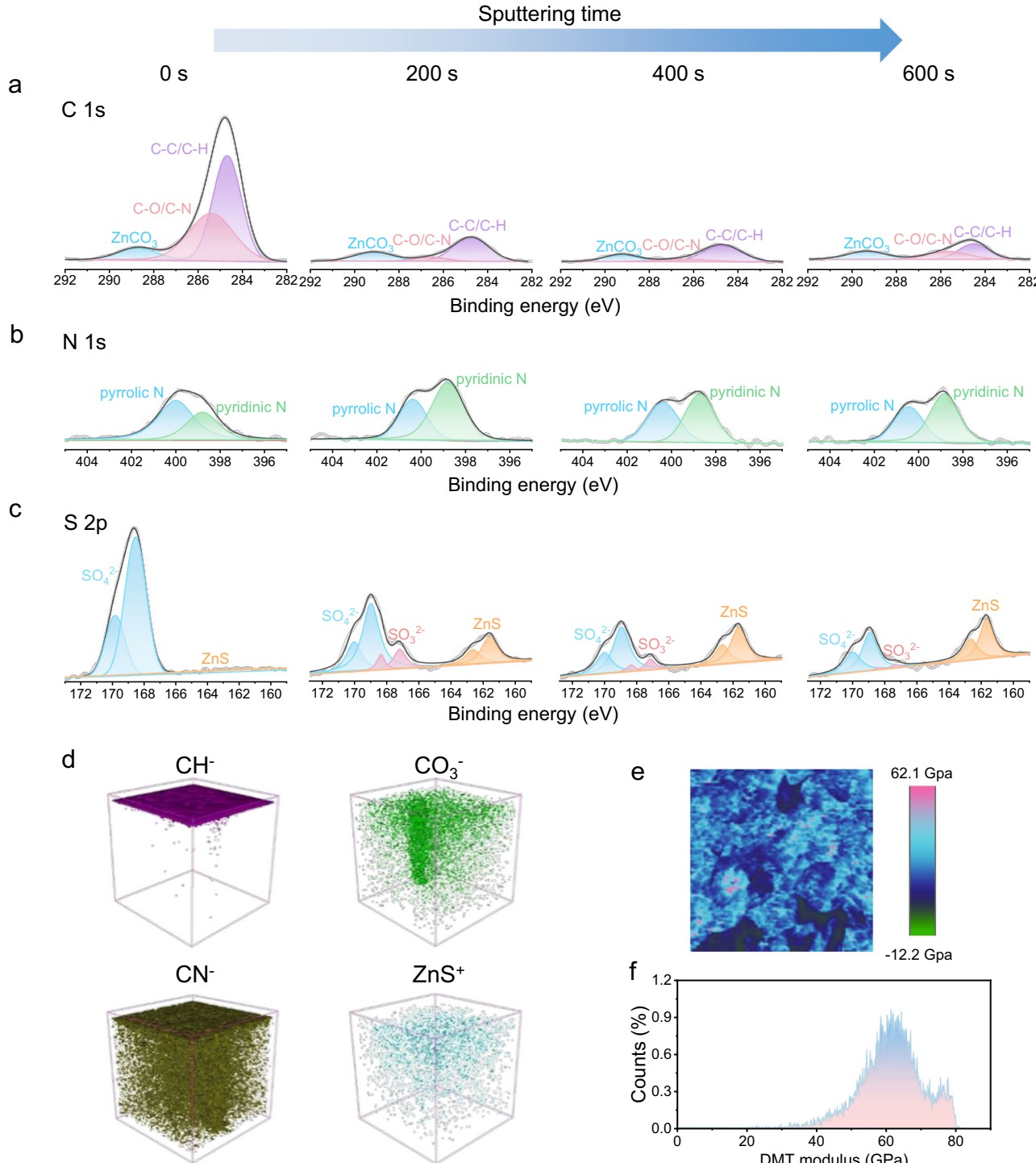

**Fig. 3 | Interfacial studies of chemical composition and mechanical stability of bilayer SEI. a–c** XPS spectra of (**a**) C 1 s, (**b**) N 1 s, and (**c**) S 2p after Ar+ sputtering from 0 s to 600 s. **d**, 3D views of organic and inorganic components in the ToF-SIMS of SEI. **e–f** DMT modulus distribution of bilayer SEI.

compositions with high viscoelasticity can accommodate large volume changes during Zn plating. The amorphous inner layer homogenizes $Zn^{2+}$ transport for homogeneous, dense Zn deposition. Consequently, it enables dendrite-free, highly reversible, and high-utilization Zn electrodes. We therefore summarize the bilayer SEI formation pathway as shown in Fig. 4b. Initially, DMI molecules undergo electrochemically induced redox reactions via the electron-driven and proton transfer mechanisms, leading to the electrochemical decomposition to yield dimethylamine and $CO_2$. Subsequently, due to the weak acidic nature of the solution, dimethylamine rapidly complexes with $H^+$ and $SO_4^{2-}$ in

the solution to form $[C_4H_{14}N_2]SO_4$. Simultaneously, the decomposed $CO_2$ dissolves in the electrolyte and reacts with $Zn^{2+}$ therein, generating $ZnCO_3$. In the meanwhile, a small portion of $SO_4^{2-}$ adsorbed on the surface of the Zn electrode undergoes reduction to $S^{2-}$ in the electron-driven and proton-driven processes[36,37]. The $S^{2-}$ further combines with $Zn^{2+}$ to form ZnS[38]. In-situ pH monitoring showed a decrease in pH at the Zn electrode during the Zn deposition process, forming a locally acidic environment, which further corroborates the formation of ZnS (Supplementary Fig. 21 and Supplementary Movies 1, 2). The formation of the bilayer SEI structure is due to the kinetics differences of the

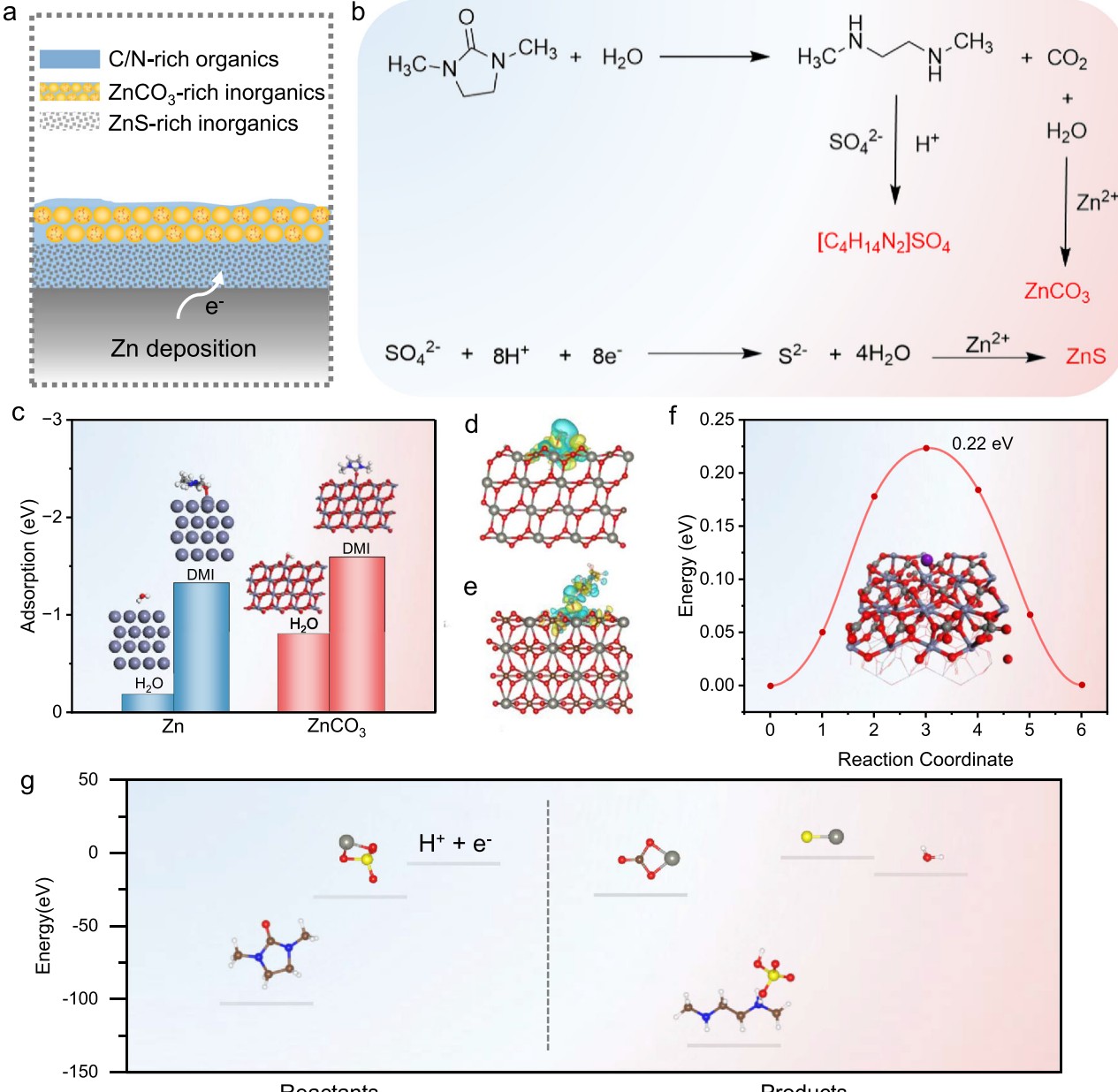

**Fig. 4 | Mechanism of bilayer SEI formation. a** Schematic presentation of the bilayer SEI. **b** Possible bilayer SEI formation pathways. **c** The adsorption energy of $H_2O$/DMI on Zn and $ZnCO_3$ surfaces. **d**, **e** Differential charge density maps of (**d**) $H_2O$ and (**e**) DMI adsorbed on $ZnCO_3$. **f** The migration energy of Zn transport through $ZnCO_3$. **g** The total energy of reactants and products in the overall formation reaction.

individual reactions. The overall reaction equation can be described as follows:

$$C_5H_{10}N_2O + 2ZnSO_4 + 8H^+ + 8e^- \rightarrow ZnCO_3 + ZnS \\ + [C_4H_{14}N_2]SO_4 + 2H_2O \quad (1)$$

Furthermore, density functional theory (DFT) calculations were carried out to validate the formation mechanism of the bilayer SEI (Supplementary Data 1). The electrostatic potential (ESP) mapping of DMI and $H_2O$ suggests that O atoms act as adsorption contact sites with the substrate (Supplementary Fig. 22)[39]. As shown in Fig. 4c, the adsorption energy of DMI on Zn is -1.33 eV, which is much stronger than that of $H_2O$. This result reveals that DMI is much easily adsorbed on the surface of Zn electrode to realize the subsequent

decomposition to generate SEI (Supplementary Figs. 23-26)[40]. DMI also has a stronger interaction with $ZnCO_3$ compared to $H_2O$ after the bilayer SEI formation, where $H_2O$ molecules can be kept relatively far away from the surface of Zn, thus inhibiting side reactions caused by $H_2O$ (Fig. 4c–e and Supplementary Figs. 27-30)[41]. We also calculated the migration energy barriers of $Zn^{2+}$ through $ZnCO_3$ by the climbing image nudged elastic band method. It is indicated that the migration energy barrier of $Zn^{2+}$ through $ZnCO_3$ is about 0.22 eV, which is conducive to rapid transport of $Zn^{2+}$ through the SEI for deposition (Fig. 4f and Supplementary Fig. 31). Finally, the total energy of the reactants and products in the reaction was qualitatively estimated by DFT calculations (Fig. 4g and Supplementary Table 1)[20]. The calculated reaction energy is -1.58 eV, which proves that the reaction is thermodynamically feasible and validates the reliability of our proposed mechanism for SEI formation.

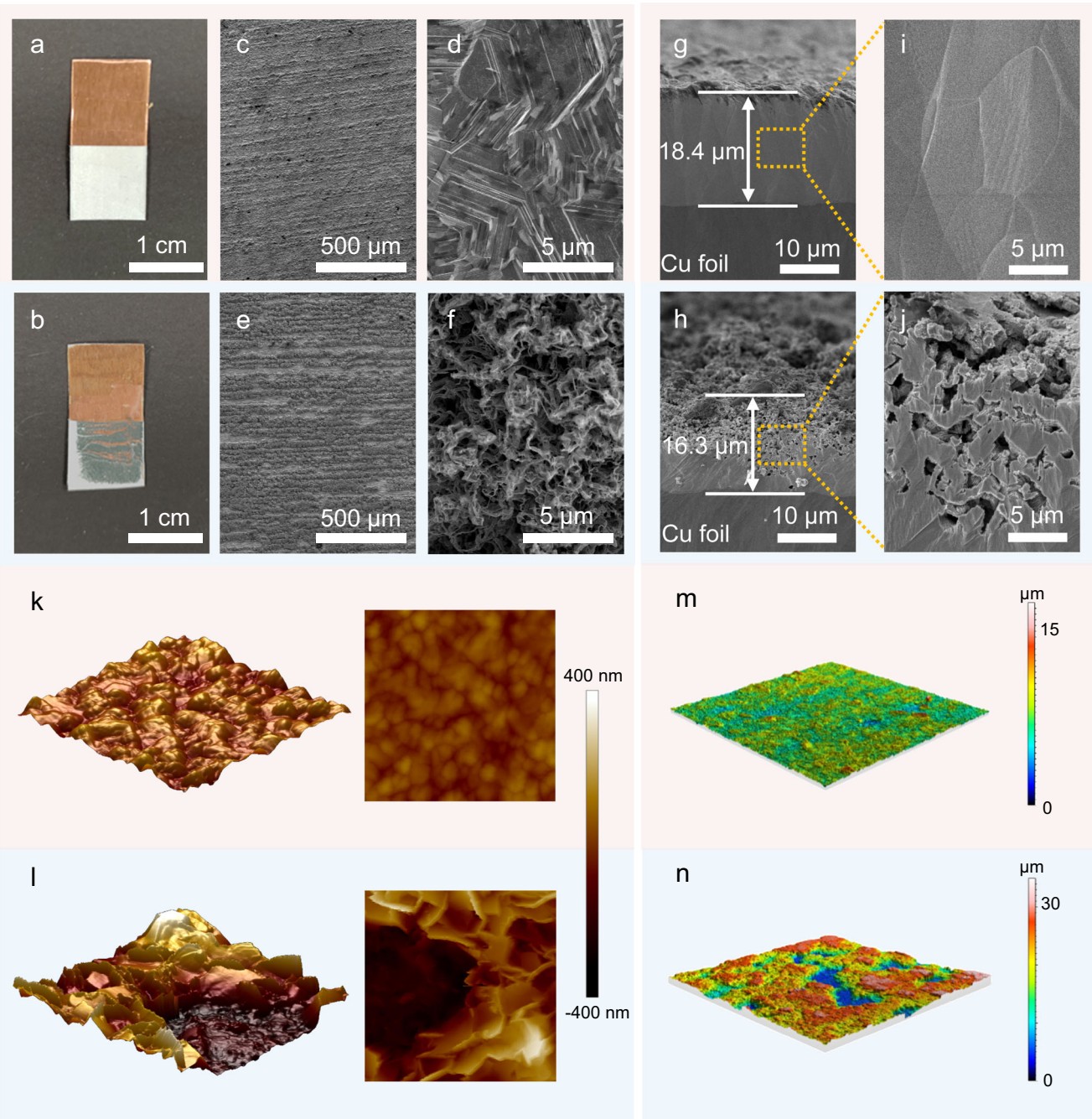

**Fig. 5 | Characterization of Zn electrodeposition morphology with/without bilayer SEI. a, b** Digital photos of Zn deposited on Cu foil (**a**) with bilayer and (**b**) without SEI. **c-f**, SEM images of Zn deposited on Cu foil (**c, d**) with bilayer and (**e, f**) without SEI. **g–j** Cross-sectional SEM images of Zn deposited on Cu foil (**g, i**) with bilayer and (**h, j**) without SEI. **k–l** AFM topography images of Zn deposited on Cu foil (**k**) with bilayer and (**l**) without SEI. **m-n**, CLSM images of Zn deposited on Cu foil (**m**) with bilayer and (**n**) without SEI.

## Morphology of Zn deposition

The effect of bilayer SEI on Zn deposition/dissolution morphology was investigated by scanning electron microscope (SEM), AFM and three-dimensional confocal laser scanning microscopy (3D CLSM)[42,43]. As shown in Fig. 5a, b, the digital photographs show that the Zn deposited at a large areal capacity of 10 mAh cm$^{-2}$ on a copper (Cu) foil with the bilayer SEI is uniform from a macroscopic point of view, while Zn deposited without the bilayer SEI is unevenly distributed. SEM images of Zn deposited on Cu with the bilayer SEI exhibit a uniform, dense, and dendrite-free morphology (Fig. 5c, d). In sharp contrast, a porous, loose and dendritic morphology of Zn deposition is obtained without

the bilayer SEI (Fig. 5e, f). In addition, the cross-sectional images of Zn deposition provide more visual evidences of the homogenization of the Zn plating. The cross-sectional morphology of Zn deposition with the bilayer SEI shows good homogeneity without dendrites, benefiting from the homogenization of Zn$^{2+}$ transport through SEI (Fig. 5g and Supplementary Fig. 32). However, a large amount of fluffy Zn grows along the vertical direction of the Cu substrate without the bilayer SEI, validating the inhomogeneous plating of Zn and the formation of dendrites without SEI (Fig. 5h and Supplementary Fig. 33). The high-resolution cross-sectional SEM image reveals that the bilayer SEI enables dense Zn deposition (actual thickness: 18.4 μm) close to the

theoretical Zn deposition thickness of 17.1 μm under 10 mAh cm⁻² (Fig. 5i)[7]. However, Zn deposition layer without SEI results in highly porous, nodular Zn morphology (Fig. 5j), with a thickness of 16.3 μm, which is much lower than the theoretical thickness due to severe side reactions during the Zn deposition process. This dense Zn deposition not only can reduce the contact area with the electrolyte, but also can maintain good electron transport and mass transport, thus reducing the generation of inactive Zn and improving the reversibility of Zn plating/stripping[44]. AFM and 3D CLSM were conducted to further confirm the differences in uniformity of Zn deposition. It displays in Fig. 5k–n that Zn deposited on Cu with the bilayer SEI has a much lower difference in height compared to that without SEI. In other words, uniform and dendrite-free Zn deposition can be achieved by the uniform $Zn^{2+}$ transport regulation of the bilayer SEI.

## Electrochemical performance of Zn electrode

Furthermore, the cycling performance of Zn electrodes in different electrolytes using Zn||Zn symmetric cells were tested under different areal capacities and current densities. As shown in Fig. 6a, the symmetric cell with bilayer SEI presents a durable cycle life of 1600 h, much

longer than that of no SEI under 1 mAh cm⁻² and 2 mA cm⁻². In order to evaluate the availability of the bilayer SEI under practical conditions, long-term stability tests were conducted under high areal capacity. The symmetric cell without SEI suffers from failure which can only run for less than 100 h under 10 mAh cm⁻² and 10 mA cm⁻² due to the formation of Zn dendrites, while the symmetric cell with the bilayer SEI demonstrates improved stability over 900 h (Fig. 6b). These results reveal that the bilayer SEI can distinctly promote the cycling performance of Zn electrodes. Linear sweep voltammetry (LSV) measurements of the Zn electrode with/without bilayer SEI were carried out (Supplementary Fig. 34). The result indicates that the HER is significantly suppressed on the bilayer SEI of Zn electrode. Moreover, the Zn plating/stripping reversibility in the two aqueous electrolyte solutions was evaluated in Cu||Zn asymmetric cells. The average CE (ACE) of Zn plating/stripping with the bilayer SEI enables an high ACE of 99.95% for more than 4800 cycles at 1 mAh cm⁻² and 2 mA cm⁻² (Fig. 6c, d). In order to promote the practical application of Zn electrodes and achieve high energy density of ZIBs, it is necessary to evaluate Zn electrodes under high ZUR. However, achieving high ZUR at high areal capacity remains highly challenging, and only a limited

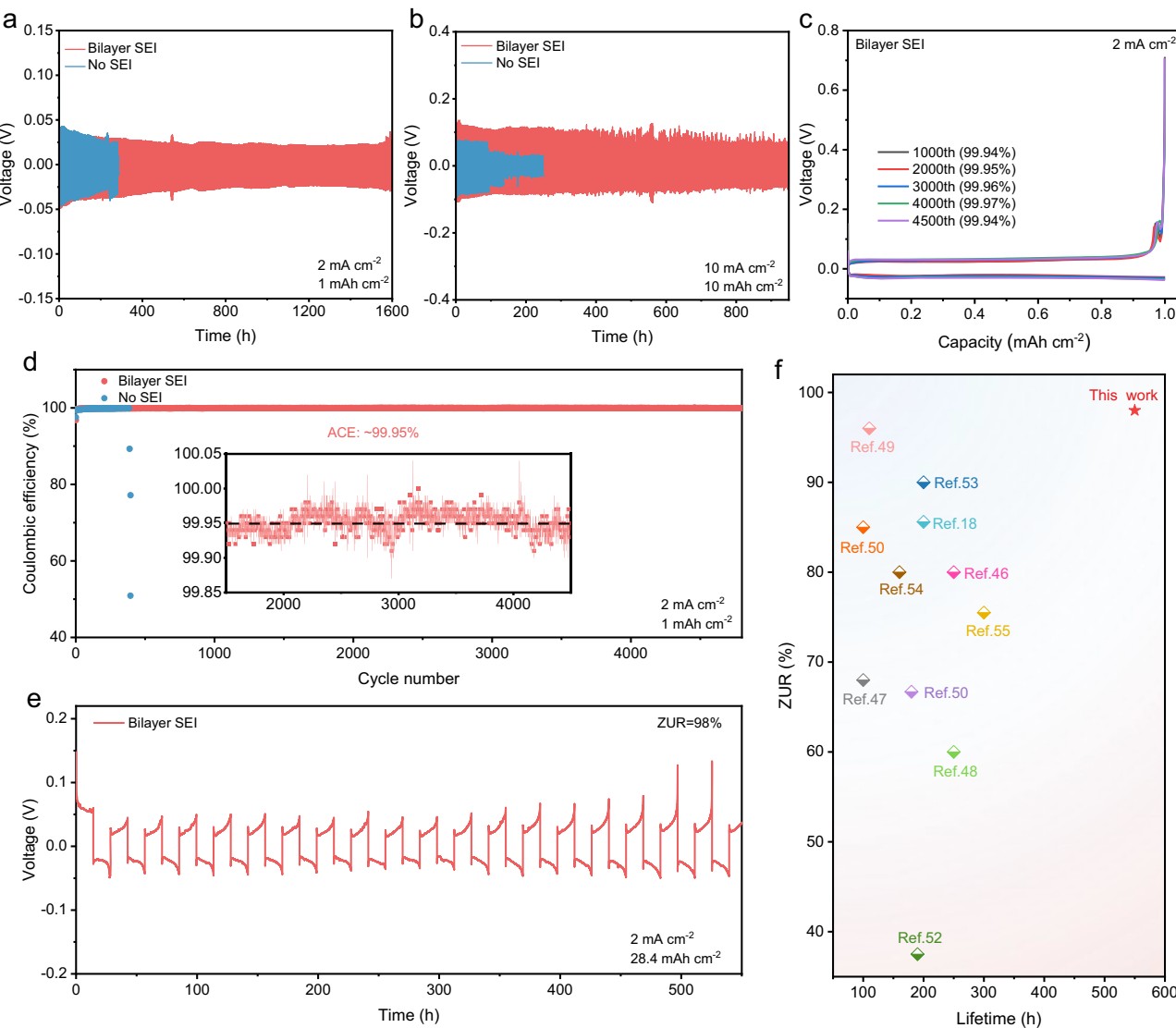

**Fig. 6 | Electrochemical performance of Zn electrode (25 °C). a** Long-term stability of symmetric cells at 1 mAh cm⁻² and 2 mA cm⁻². **b** Long-term stability of symmetric cells at 10 mAh cm⁻² and 10 mA cm⁻². **c** GCD curves of Zn plating/stripping in Cu||Zn asymmetric cells at 1 mAh cm⁻² and 2 mA cm⁻² during cycling. **d** The CE of Zn plating/stripping in Cu||Zn asymmetric cells for over 4800 cycles at 1 mAh cm⁻² and 2 mA cm⁻². **e** Long-term stability of high ZUR symmetric cells at 28.4 mAh cm⁻² and 2 mA cm⁻². **f** Comparison of ZUR and lifetime between our work and previous studies on high ZUR.

number of related studies have been reported[45]. When the ZUR increases to as high as 98%, the Zn‖Zn symmetric cell with the bilayer SEI still shows stable plating/stripping for more than 550 h under a high areal capacity of 28.4 mAh cm$^{-2}$, which is a high value under the conditions of such a high ZUR and high areal capacity (Fig. 6e). These results show that Zn electrode with high ZUR and CE can be achieved, benefiting from the bilayer SEI to promote uniform Zn deposition and reversibility of the Zn deposition/dissolution (Supplementary Fig. 35). We have summarized studies on high ZUR in literature and compared them with our work[46–56]. The optimal Zn‖Zn symmetric cells can only achieve stable cycling less than 200 h in previous studies when the ZURs are over 90% (Fig. 6f and Supplementary Table 2). However, in this study, the Zn‖Zn symmetric cells exhibited stable cycling for over 550 h even under a ZUR as high as 98%.

## Electrochemical performance of full cells with high ZURs

In order to verify the feasibility of the bilayer SEI we constructed in practical ZIBs, we assembled different types of full cells under

controlled N/P ratios with high ZURs. As shown in Fig. 7a, the Zn‖ active carbon (Zn‖AC) full cell based on bilayer SEI can remain stable cycling performance for more than 9000 cycles with the capacity retention of 80% at the specific current of 1 A g$^{-1}$ (N/P = 2). Nevertheless, the capacity of the Zn‖AC full battery without the bilayer SEI decays about 50% in 3000 cycles. The charge/discharge curves at different cycling numbers reflect the significantly improved cycling performance of the Zn‖AC full cell with the bilayer SEI under a high ZUR of 50% (Fig. 7b and Supplementary Fig. 36). Furthermore, we assembled Zn‖iodine (Zn‖I$_2$) full cell in different electrolytes at a lower N/P ratio of 1.5. Zn‖I$_2$ full cell based on the bilayer SEI exhibits stable cycling performance over 5000 cycles at 2.1 A g$^{-1}$ with negligible capacity decay under a high ZUR of 67%, while Zn‖I$_2$ full cell without SEI rapidly fails after less than 100 cycles (Fig. 7c, d and Supplementary Fig. 37). In order to further enhance the ZUR of ZIBs, we assembled initially anode-free Zn‖bromine (Zn‖Br) pouch cell (5*5 cm, 1 positive electrode and 1 negative electrode) without any pre-stored Zn (Supplementary Fig. 38). The Zn‖Br pouch cell with the bilayer SEI can

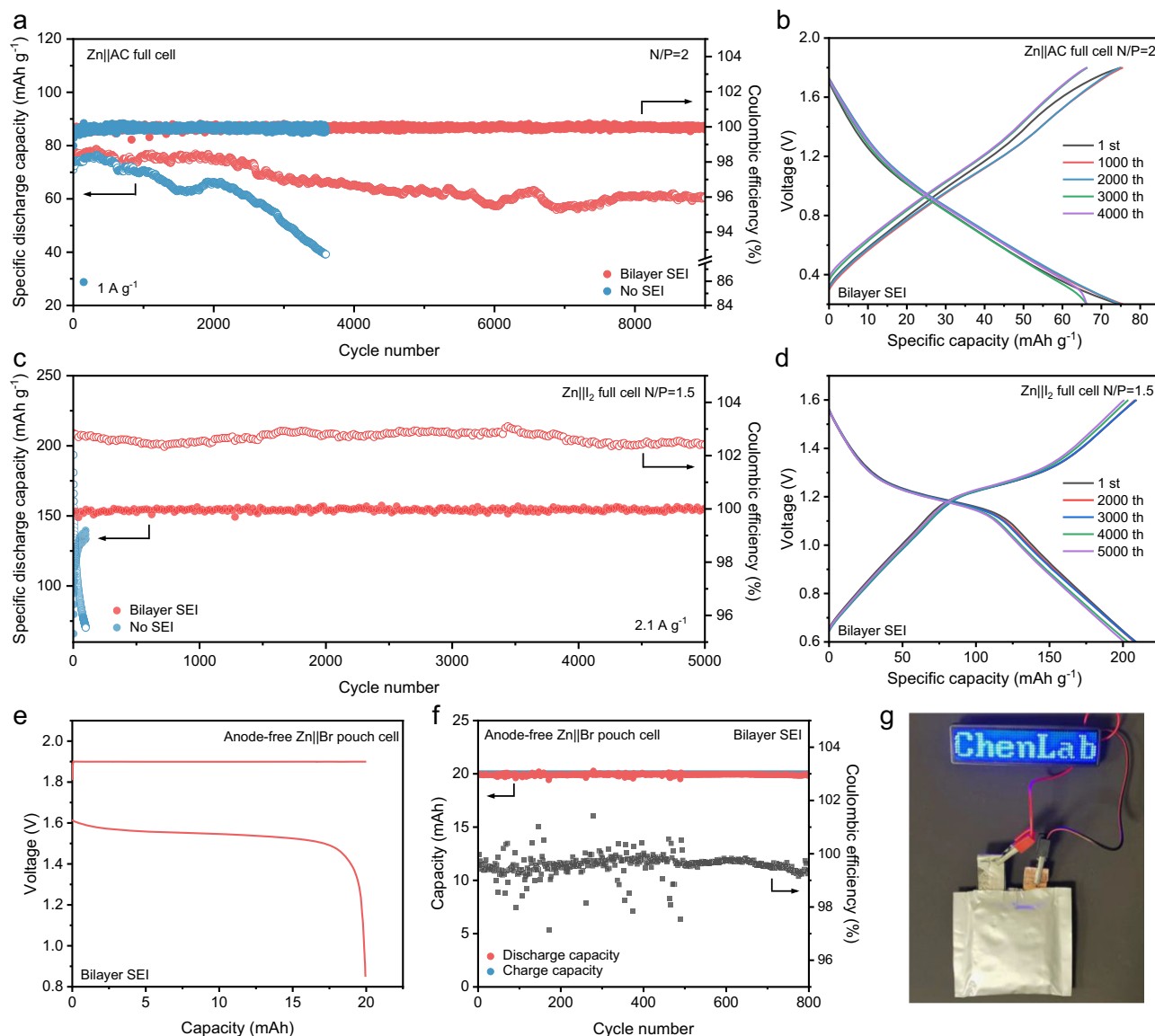

**Fig. 7 | Electrochemical performance of Zn full cells with low N/P ratios and high ZURs (25 °C). a** Long-term stability of Zn‖AC full cells with/without the bilayer SEI at N/P of 2. **b** GCD curves of Zn‖AC full cell with the bilayer SEI at N/P of 2. **c** Long-term stability of Zn‖I$_2$ full cells with/without the bilayer SEI at N/P of 1.5. **d** GCD curves of different cycles of Zn‖I$_2$ full cell with the bilayer SEI at N/P of 1.5.

**e** GCD curve of anode-free Zn‖Br pouch cell with the bilayer SEI (The discharge current is 40 mA). **f** Long-term stability of anode-free Zn‖Br pouch cell with the bilayer SEI (The discharge current is 40 mA). **g** An LED digital screen powered by anode-free Zn‖Br pouch cell with the bilayer SEI.

maintain durable stability for more than 300 cycles under the capacity of 20 mAh at 40 mA (Fig. 7e, f) without cell swelling. We also demonstrate that Zn||Br pouch cell can successfully light up an LED digital screen (Fig. 7g). These demonstrations of different full cells show that the bilayer SEI remains effective under high ZUR, highlighting the potential of the bilayer SEI strategy to Zn electrodes for practical battery applications.

## Discussion

In summary, we have designed a robust bilayer SEI that simultaneously achieved the homogeneous $Zn^{2+}$ transport and mechanical stability for high Zn utilization rate (ZUR) and Coulombic efficiency (CE) of Zn electrode. This bilayer SEI on Zn surface consisted of a crystalline $ZnCO_3$-rich outer layer and an amorphous ZnS-rich inner layer. The ordered outer layer could improve the mechanical stability during repeated Zn plating/stripping process, and the amorphous inner layer could homogenize $Zn^{2+}$ transport for homogeneous, dense Zn deposition. Compared to regular single-layer SEI, the crystalline/amorphous bilayer SEI was much more homogeneous and robust, which was durable to suppress the fracture of SEI during cycling. Therefore, Zn plating/stripping with the bilayer SEI could achieve high CE of 99.95% (1 mAh cm$^{-2}$ and 2 mA cm$^{-2}$) and ZUR of 98% (28.4 mAh cm$^{-2}$ and 2 mA cm$^{-2}$). Moreover, three different categories of Zn full batteries with the bilayer SEI all exhibited improved cycling stability at high ZURs.

## Methods

### Materials

All the reagents and chemicals used in this work were commercially available: zinc sulfate heptahydrate ($ZnSO_4·7H_2O$, Sinopharm Chemical Reagent Co., Ltd, 99.5%), 1,3-Dimethyl-2-imidazolidinone (DMI, Aladdin, 99%), sodium sulfate ($Na_2SO_4$, Sinopharm Chemical Reagent Co., Ltd, 99%), zinc foil (Zn, 50 μm, Sinopharm Chemical Reagent Co., Ltd, 99.9%), copper foil (Cu, Sinopharm Chemical Reagent Co., Ltd, 99.9%), activated carbon (AC, Fuzhou Yihuan Carbon., Ltd, 99%, the particle size of activated carbon is about 9 μm and the porosity is about 40%), iodine ($I_2$, Sinopharm Chemical Reagent Co., Ltd, 99.8%), tetrapropylammonium bromide (TPABr, Aladdin, 98%), titanium foil (Ti, Beijing Xinruichi Technology Co., Ltd., 99.9%), and N-Methylpyrrolidone (NMP, Aladdin, 99%), polyvinylidene fluoride (PVDF, Arkema, 99%), conductive carbon black (AB, BLACK PEARLS, 99.9%, the particle size of activated carbon is about 15 nm and the porosity is about 85%.), Super Conducting Carbon Black (Super P, TIMCAL, 99.9%, the particle size of activated carbon is about 45 nm and the porosity is about 80%.), carboxymethyl cellulose (CMC, Kermel, 99%, the viscosity of CMC is between 1200 and 1600 mPa/s), carbon felt (CF, Tan Qian Lang Materials, 98%, the thickness of CF is about 3 mm and the porosity is about 90%.), and deionized water (resistance of 18.2 MΩ, Milli Q).

### Physicochemical characterizations

The solvation structures of electrolytes were studied by Raman spectroscopy (Raman, LabRAM HR Evolution) with a wavelength of 532 nm and fourier transform infrared spectroscopy (FTIR, Nicolet Impact 410 FTIR Infrared Instrument). Transmission electron microscope (TEM, JEM-F200) and aberration-corrected transmission electron microscope (FEI Themis G2) were carried out to analyze the nanostructure and components of SEI on Zn surface. Regarding the preparation method of TEM samples, we employed an in-situ electro-deposition approach. A Cu TEM grid was clamped with a reverse clamp and immersed in the electrolyte as the working electrode, with Zn foil serving as the counter electrode and reference electrode. Zn was then electro-deposited in-situ onto the Cu grid. After deposition, the Cu grid was immediately rinsed with water and ethanol. The chemical components of bilayer SEI on Zn surface were investigated by X-ray

photoelectron spectroscopy (XPS, Kratos Axis supra$^+$) with Al K$_\alpha$ source under Ar$^+$ sputtering at 2 kV for 0, 200, 400, and 600 s. Time-of-flight secondary ion mass spectrometry depth profiling data were collected using a PHI nano TOF II instrument (TOF-SIMS, ULVAC PHI, Japan) equipped with a bismuth liquid metal ion gun and Ar sputtering gun. The analysis beam for this study was generated by the liquid metal ion gun, specifically a 30 keV bismuth source, utilizing a Bi-cluster liquid metal ion gun over an area of 100 μm × 100 μm. The target current was measured as 2 nA. A 3 kV 100 nA Ar source was employed to etch through the sample over an area of 400 μm × 400 μm. Atomic force microscopy (AFM, Bruker, Demension Icon) was used to analyze the mechanical stability of bilayer SEI by using an AFM probe (Bruker, f$_0$ = 525 kHz, k = 200 N m$^{-1}$). Cold field emission scanning electron microscope (SEM, Hitachi SU8220) was used to investigate the surface and cross-sectional morphology of Zn deposition in different electrolytes. The height difference of Zn deposited with/without bilayer SEI was characterized via confocal laser scanning microscopy (CLSM, Zeiss LSM900). The ionic conductivity and viscosity of electrolytes were determined using a conductivity meter (Shanghai Yueping, DDS-11A) and Rheometer (Shanghai Yixin Scientific Instrument Co., Ltd. NDJ-8S5S9S) at 25 °C, respectively.

### Preparation of AC positive electrode

Commercial AC powder, AB, and PVDF binder with a mass ratio of 8:1:1 (the mass percentage of active material AC is about 80%) were dispersed in NMP to form a homogeneous slurry. The slurry was then coated onto a Ti foil via the doctor blading, and dried at 80 °C for at least 24 h, subsequently. The mass loading of AC is about 3–4 mg cm$^{-2}$.

### Preparation of $I_2$ positive electrode

$I_2$ and AC were mixed according to a mass ratio of 1:4, and then transferred to a hydrothermal kettle to react at 80 °C for 4 h. After natural cooling, $I_2$@AC powder was obtained. Afterwards, $I_2$@AC, super P and CMC binder with a mass ratio of 8:1:1 (the mass percentage of active material $I_2$ is about 20%) were dispersed in $H_2O$ to form a homogeneous slurry and then coated onto a Ti foil via the doctor blading, and dried at 40 °C for 12 h, subsequently. The mass loading of $I_2$ is 0.6–0.7 mg cm$^{-2}$.

### Preparation of Br-based positive electrode

In the preparation of the Br-based positive electrodes, tetrapropylammonium tribromide (TPABr$_3$), TPABr, conductive carbon black, and PVDF were combined in a weight ratio of 6:2:1:1 (the mass percentage of active material TPABr$_3$ is about 60%), and mixed with NMP to create a slurry. It was then dropped onto a carbon felt and dried at 80 °C for 12 h. The active materials of Br-based positive electrode loading is about 25–30 mg cm$^{-2}$.

### Preparation of electrolyte

Taking 100 mL electrolyte as an example, for 2 M $ZnSO_4$ we weigh 57.508 g $ZnSO_4·7H_2O$, add it to a beaker, add appropriate amount of water to dissolve it completely, transfer it to a volumetric flask, and then dilute it to 100 mL. As for 2 M $ZnSO_4$ + 10 mM DMI, we weigh 57.508 g $ZnSO_4·7H_2O$ and 0.1141 g DMI separately, add them to a beaker, add appropriate amount of water to dissolve it completely, then transfer it to a volumetric flask and dilute it to 100 mL.

### Electrochemical tests

All of the electrochemical tests were conducted in air at temperature of 25 °C. The symmetric cells, asymmetric cells, Zn||AC full cells and Zn||$I_2$ full cells were conducted in coin-type cells (CR2032) with 75 μL 2 M $ZnSO_4$ or 2 M $ZnSO_4$ + DMI electrolytes. The dimensions of all electrodes we used in coin-type cells are circular with a diameter of 12 mm. The glass fiber separators we used were 16 mm in diameter and 675 μm in thickness. The symmetric cells were assembled via Zn foils as

working electrode and counter electrode, simultaneously. The asymmetric cells were assembled via Cu as the working electrode and Zn as the counter electrode (The cut-off voltage is 0.7 V). Prior to assembling Zn||Zn symmetric cells with high ZUR, the Zn foil was firstly cleaned with deionized water and anhydrous ethanol. The surface was then gently polished with fine sandpaper to remove the oxide layer and impurities, followed by further cleaning with water and anhydrous ethanol. The treated Zn foil was cut into 12 mm diameter discs and weighed. The actual capacity of the Zn negative electrode used was calculated based on its weight and the theoretical specific capacity of Zn (820 mAh g$^{-1}$). At this point, the ZUR is defined as the design cycling capacity divided by the actual capacity of the Zn negative electrode used. Assuming the weight of the Zn negative electrode is $x$ g and the cycling capacity is $y$ mAh, the ZUR can be calculated using the formula: ZUR=$y$/($x$*820). The negative electrodes of full cells at low N/P ratios were prepared by pre-depositing a certain amount of Zn on Cu foils. Regarding the calculation method of the N/P ratio in this work, we computed the theoretical capacity of the positive electrode. Then, a predetermined amount of Zn was pre-deposited on the Cu foil for the negative electrode to achieve the set N/P ratio. Specifically, we weighed the mass of the active material in the positive electrode and calculated the theoretical capacity based on its theoretical specific capacity. Assuming an N/P ratio of 2, the capacity of Zn pre-deposited on the Cu foil for the positive electrode would be twice that of the corresponding positive electrode. For the Zn||AC battery, the cut-off voltage was 1.8 V. For the Zn||I$_2$ battery, the cut-off voltage was 1.6 V. Anode-free Zn||Br full cells were assembled by Cu foil as the current collector of negative electrode and Br-based positive electrode with 2 M ZnSO$_4$ + 10 mM DMI + 0.5 M ZnBr$_2$ electrolyte in pouch cells. The pouch cell consists of a pair of single-layer positive and negative electrodes. The size of positive and negative electrodes was 4 cm × 4 cm. The size of the separator was 5 cm × 5 cm. The amount of electrolyte was controlled to just infiltrate the separator. The depth of discharge (DoD) of the Br positive electrode was ~50%. The discharge current is 40 mA. The cycling performance and galvanostatic charge-discharge were tested on LandHe CT3002A (Wuhan, China) and Neware CT4008Tn (Shenzhen, China) battery test systems. The linear sweep voltammetry (LSV) was tested on Biologic VMP3 multi-channel electrochemical workstation (France). For HER test, the bare Zn foil and Zn foil with bilayer SEI (after cycling in 2 M ZnSO$_4$ + 10 mM DMI electrolyte) were separately used as the working electrode (1 cm × 1 cm), Pt as counter electrode (1 cm × 1 cm) and Ag/AgCl as reference electrode in 2 M Na$_2$SO$_4$ electrolyte (20 mL). For the electrochemical tests conducted with coin cells, the number of samples for each test is at least 10. For the three-electrode test and the pouch cell test, we conduct at least 3 samples for each test.

### Computational method

The Vienna ab initio Simulation Package (VASP) was used in this work[57,58]. The Perdew–Burke–Ernzerhof (PBE)[59] functional within the generalized gradient approximation (GGA) was chosen to describe the electronic exchange and correlation interactions. And the projected augmented wave (PAW) method[60,61] was taken valence electrons into account using a plane wave basis set with a kinetics energy cutoff of 450 eV. Partial occupancies of the Kohn−Sham orbitals were allowed to use the Gaussian smearing method and a width of 0.1 eV. The electronic energy was considered self-consistent when the energy change was smaller than $10^{-5}$ eV. A geometry optimization was considered convergent when the energy change was smaller than -0.02 eV/Å. This slab was separated by a 15 Å vacuum layer in the $z$ direction between the slab and its periodic images. A single gamma-point k-point grid for Brillouin zone was used for geometry optimization and 2 × 2 × 1 k-point for electrostructure calculation. Grimme's DFT-D3 methodology[62] was used to describe the dispersion interactions among all the atoms in adsorption models of interest.

Transition states were searched by climbing image nudged elastic-band method (CI-NEB)[63].

The adsorption energy (E$_{ads}$) of an adsorbate A was defined as:

$$E_{ads} = E_{A/surf} - E_{surf} - E_A$$

where E$_{A/surf}$, E$_{surf}$ and E$_A$ are the energy of A (H$_2$O, DMI, Zn) adsorbed on the substrates surface, the energy of substrates clean surface, and the energy of A in the gas phase, respectively.

## Data availability

The data supporting the findings of this study are available in the Methods section of the article and in the related supplementary information. Source data are provided with this paper.

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

## Acknowledgements

W.C. acknowledges the funding support from the National Natural Science Foundation of China (52471242, 92372122) and the Fundamental Research Funds for the Central Universities (Grant KY2060000150, GG2060127001, WK2060000040). Z.L. acknowledges the National Natural Science Foundation of China with Grant No. 22203086, and the project funded by the China Postdoctoral Science Foundation with Grant No. 2022M713032. Z.L. acknowledges the funding support from National Natural Science Foundation of China (22393913) and the Strategic Priority Research Program of the Chinese Academy of Sciences (XDB0450101). Y.W. acknowledges the support from NSFC No.22250004, GD BABRF No. 2022A1515010902, GZ BABRF No. 2023A04J1356, PCOSS Open Project No.202202, and National Key R&D Program of China No. 2023YFA1509001. We thank the support from USTC Center for Micro and Nanoscale Research and Fabrication and NEWARE. This work was partially carried out at the Instruments Center for Physical Science, University of Science and Technology of China. This work was also supported by the advanced computing resources provided by the Supercomputing Center of the USTC.

## Author contributions

Y.M., M.W., and J.W. contributed equally to this work. W.C. conceptualized the concept and supervised this work. W.C., Y.M. designed the detail experiments project. Y.W., X.H., and X.Z. contributed the TEM test. Z.L. and J.W. contributed DFT simulations. Y.M. conducted the electrochemical performance of half cells and full cells. M.W. helped with the characterization of materials. Y.M. analyzed the data of electrochemical test and characterization. M.S., Z.X., R.L., Z.Zhu., K.N., and Z.Zhang. assisted in analyzing the data. W.C. and Y.M. wrote the paper. All authors discussed the results and commented on the manuscript.

## Competing interests

The authors declare no competing interests.
