## [Peer Review File · Nature Communications]

Robust Bilayer Solid Electrolyte Interphase for Zn Anode with High Utilization and EfficiencyREVIEWER COMMENTS

Reviewer #1 (Remarks to the Author):

This manuscript provides a comprehensive analysis of the bilayer SEI (with ZnCO₃-rich outer layer and ZnS-rich inner layer) formed by the electrolyte additive (DMI) in zinc-based batteries.

Previous studies have focused on modifying the SEI using electrolyte additives to achieve reversible Zn plating/stripping behavior. Also, compared to the study on single-layer SEI, the investigation of bilayer SEI remains limited. In particular, a paper using DMI as an aqueous zinc battery electrolyte additive has already been published, but an in-depth analysis of SEI was insufficient.

This study is anticipated to enhance our understanding of SEI formation mechanisms influenced by electrolyte additives and further provide a basis for studies to overcome the limitations of existing SEI. Moreover, its utilization with various cathodes is expected to propel the advancement of zinc-based batteries.

For these reasons, I recommend the publication of this research in Nature Communications after addressing the following comments.

Detailed comments:

1. Errors in Main Manuscript

1) On page 7, “(Figure 2j-2l)” should be corrected to “(Fig. 2j-2l)” and made bold to ensure the consistency in notation.

2) On page 14, It seems there might be a discrepancy between the text and the figures. It appears that Fig. 5g and Fig. 5h are the relevant figures for checking thickness. Therefore, content adjustment is necessary to ensure accuracy. Additionally, further analysis and interpretation should be provided for Fig. 5i and Fig. 5j.

2. Figure Correction

- 1) On page 13, the description for Fig. 4d and 4e should be corrected from “absorbed” to “adsorbed”.
- 2) On page 15, it would be helpful to specify the observed values of SEI thickness on Fig. 5g and Fig. 5h.
- 3) On page 15, since SEI is commonly found in current batteries, it might be prudent to consider using 'without bilayer SEI' instead of 'without SEI' in Fig 5. Additionally, 'bare SEI' appears to be a suitable alternative term.
- 3) On page 17, As previously mentioned, it would be advisable to replace 'without SEI' in Fig. 6 with a more appropriate term such as 'bare SEI', or 'without bilayer SEI'.

4) More details

- 1) On page 9, the reference of the sentence “Pyridinic C-N species are formed through the electrochemical decomposition of DMI and pyrrolic C-N species are associated with DMI adsorption.” is required.
- 2) On page 9, the interpretation of the presence of SO_4^{2-} and SO_3^{2-} ions in the outer layer depicted in Fig. 3c appears to be insufficient. If their presence is attributed to the ZnSO_4 electrolyte, it would be beneficial to explicitly mention it for clarification.
- 3) On page 14, a reference is needed for the statement regarding the “theoretical Zn deposition thickness of $17.1 \mu\text{m}$ ”.

5) Additional experiment

- 1) On page 6, Fig. 2a was noted the absence of a distinct interfacial layer, but the explanation provided remains inadequate. Further investigation into the bare interfacial layer's structure via EDS mapping is imperative for a comprehensive understanding.
- 2) On page 12, Further analysis is required to contextualize the significance of the value (0.22 eV) mentioned in Fig. 4f. It would be advantageous to compare it with the migration energy barrier of the SEI layer in additive-free aqueous zinc battery for a more comprehensive understanding.
- 3) To demonstrate the effect of the unique bilayer structure in this experiment, it would be beneficial to include experiments comparing with Zn cells having single-layer SEI. When

compared with Zn cells without SEI layer, the difference seems to focus more on the presence or absence of SEI rather than the effect of the bilayer structure.

6) Supporting information

1) Adding a brief explanation in Supplementary Fig. 2 elucidating the reason behind the performance decline with increasing DMI concentration would be beneficial.

Reviewer #2 (Remarks to the Author):

In this manuscript, the authors use 1,3-dimethyl-2-imidazolidinone (DMI) as an electrolyte additive for aqueous zinc-ion batteries. However, previous research has reported the use of DMI as an electrolyte additive for aqueous zinc-ion batteries (DOI: 10.1016/j.cej.2023.141287). Even this study provides new insights into the SEI formed by DMI, it is lack of novelty. Here, DMI induced a robust amorphous/crystalline bilayer SEI that achieves the desirable homogeneous Zn²⁺ transport and mechanical stability to achieve record ZUR and CE. The full cells based on the bilayer SEI anode show better cycling performance under low N/P ration. Several crucial issues need further exploration by the authors before next submission.

1. Why does the concentration of DMI have such a significant impact on battery performance? As depicted in Supplementary Figure 2, the performance of cells using 50 mM and 100 mM DMI is considerably worse than those using 10 mM. Consequently, exploring concentrations lower than 10 mM may also be worthwhile.

2. Is the CO₂ produced by the decomposition of 10 mM DMI sufficient to form a uniform ZnCO₃ layer? The solubility of CO₂ is known to be around 1.49 g L⁻¹ in low concentration aqueous electrolytes under ambient temperature and pressure, similar to that in water. However, ZnCO₃ is not typically observed on the surface of the zinc metal anode in conventional electrolytes. The author should give further explanation on this problem, it is better to provide some calculation formulas to prove it.

3. Regarding ZnS in the SEI layer, the authors attribute it to the further reaction of S²⁻, which is the product after the reduction of adsorbed SO₄²⁻. However, the authors do not provide

a detailed explanation. I recommend that the authors find sufficient supporting evidence, such as references, or demonstrate the plausibility of this reaction through Density Functional Theory (DFT) calculations. In addition, SO_4^{2-} is also adsorbed on the surface of the zinc metal anode in conventional electrolyte. Why is ZnS not observed in conventional electrolyte?

4. Unlike Supplementary Figure 28, Supplementary Figure 29 shows a significant presence of Cu elements distributed in the surface layer of the sample. I recommend that the authors re-test to verify this issue.

5. There appear to be some anomalies in Fig. 5I, notably a clear horizontal line in the middle of the image, which may be caused by a sudden change in the height of the sample. I suggest that the authors change the X-axis scanning direction of AFM and re-test the sample.

6. Based on Fig. 2d-I, it is challenging to conclude that the bilayer SEI is composed of the crystalline ZnCO_3 -rich outer layer and amorphous ZnS-rich inner layer, while its overall components are dominated by C/N-rich organics. Could the authors provide more images to support this conclusion?

7. On page 16, the author states, "...Linear sweep voltammetry (LSV) curves of the Zn anode with/without bilayer SEI were measured (Supplementary Fig. 30). The result indicates that the HER is significantly suppressed on the bilayer SEI of Zn anode."

According to the reaction equation for SEI formation: $\text{C}_5\text{H}_{10}\text{N}_2\text{O} + 2\text{ZnSO}_4 + 8\text{H}^+ + 8\text{e}^- \rightarrow \text{ZnCO}_3 + \text{ZnS} + [\text{C}_4\text{H}_{14}\text{N}_2]\text{SO}_4 + \text{H}_2\text{O}$, both ZnCO_3 and ZnS come from ZnSO_4 . However, in Supplementary Fig. 30, the LSV curves of Zn were obtained from 2 M Na_2SO_4 and 2 M $\text{Na}_2\text{SO}_4 + 10$ mM DMI, respectively. That is to say, there is no ZnSO_4 in these electrolytes, so what is the composition of the SEI? It shouldn't be NaCO_3 or Na_2S because they can dissolve in water.

8. In this manuscript, the I2 cathode were prepared by mixed I2 and AC according to a mass ration of 1:4. Since AC has a high content, the authors should demonstrate the contribution of AC to specific capacity of the cathode.

9. The authors should provide more experiment details, such as the TEM sample preparation process, the electrolytes of Zn-Br₂ full cells, and calculation method of N/P ratio.

Reviewer #3 (Remarks to the Author):

Meng and coworkers reported a robust bilayer SEI for protecting zinc anode, which features homogeneous Zn²⁺ transport, excellent mechanical rigidity, record Zn utilization rate, and high coulombic efficiency. Moreover, the assembled Zn-Zn symmetric cells show excellent lifetime with a high ZUR and the Zn full cell exhibit superior electrochemical stability at a low N/P ratio. Different from conventional single-layer SEIs that suffer from easily rapture and failure, this zinc anode with unique bilayer SEI with bilayer design provides a promising alternative solution for addressing the key challenges faced by the anode of aqueous zinc batteries. This work is interesting and well-organized, and the given characterization results supports their hypothesis. Although this work has many interesting results, I still have some following concerns.

1. As there are so many reported works about the in-situ forming SEI for zinc anode by introducing electrolyte additives, what are the advantages of your 1,3-Dimethyl-2-imidazolidinone (DMI) additives? A comparison table in terms of price, toxicity, viscosity, dielectric constant and other parameters is needed to highlight the advantages of this electrolyte additives.
2. Is it possible to regulate the thickness or composition of this bilayer SEI? This may further improve the final electrochemical performances.
3. According to the proposed bilayer-SEI formation mechanism, the H⁺ ions will be consumed and the pH of electrolyte will change. This will promote the zinc dendrites growth. So, the pH variations during electrochemical process will be very important. Could do please provide related data and give corresponding explanation?
4. The authors states that “The bilayer SEI 319 enables dense Zn deposition (actual thickness: 18.4 μm) close to the theoretical Zn deposition thickness of 17.1 μm.” Could do please give some explanation why the actual thickness of zinc deposition is larger than theoretical one?
5. In the 359th line, you don't need to define the ZUR for the second time. Moreover, the record zinc utilization of zinc anode with bilayer SEI is one of hightlights in this work. The detailed test and calculation methods should be given in the Methods part. The thickness and size of used zinc foil should be involved in Methods part.

Response to Reviewers' Comments

We thank the reviewers for their valuable comments, which we find very constructive and valuable. We have addressed all the comments from the reviewers to the largest possible extent. For ease of reference, the reviewers' comments and suggestions are reproduced in **blue**, our responses are in **black**, and the resulting changes to the manuscript and supporting information are in *Italicized red*. All changes have been highlighted in **yellow background** in the revised manuscript and supporting information.

Reviewer #1 (Remarks to the Author):

General Comment: This manuscript provides a comprehensive analysis of the bilayer SEI (with ZnCO₃-rich outer layer and ZnS-rich inner layer) formed by the electrolyte additive (DMI) in zinc-based batteries. Previous studies have focused on modifying the SEI using electrolyte additives to achieve reversible Zn plating/stripping behavior. Also, compared to the study on single-layer SEI, the investigation of bilayer SEI remains limited. In particular, a paper using DMI as an aqueous zinc battery electrolyte additive has already been published, but an in-depth analysis of SEI was insufficient. This study is anticipated to enhance our understanding of SEI formation mechanisms influenced by electrolyte additives and further provide a basis for studies to overcome the limitations of existing SEI. Moreover, its utilization with various cathodes is expected to propel the advancement of zinc-based batteries. For these reasons, I recommend the publication of this research in *Nature Communications* after addressing the following comments.

Our response: We are grateful to the reviewer's comments and suggestions, which are beneficial to the improvement of our manuscript. We have tried our best to conduct additional experiments and revise our manuscript according to the reviewers' comments. All changes have been highlighted in yellow background in the revised manuscript and supporting information. We sincerely hope that the revised manuscript is suitable for publication in *Nature Communications*.

Comment 1: Errors in Main Manuscript

1) On page 7, “(Figure 2j-2l)” should be corrected to “(Fig. 2j-2l)” and made bold to ensure the consistency in notation.

2) On page 14, It seems there might be a discrepancy between the text and the figures. It appears that Fig. 5g and Fig. 5h are the relevant figures for checking thickness. Therefore, content adjustment is necessary to ensure accuracy. Additionally, further analysis and interpretation should be provided for Fig. 5i and Fig. 5j.

Our response: Thanks for the reviewer’s careful reading and valuable comments. We sincerely apologize for the errors in main manuscript. We have made the necessary corrections according to the reviewers’ comments in the revised manuscript.

1) Based on the reviewer’s suggestion, we have amended “**Figure 2j-2l**” to “**Fig. 2j-2l**” and set the font to bold to ensure consistency throughout the manuscript.

2) In response to the reviewer's comments, we have added annotations for thickness in Fig. 5g-5h to ensure clarity and accuracy in image description. For the reviewers’ convenience, we put these results in **Fig. R1**. Additionally, we have provided a more detailed analysis and explanation for Fig.5g-5j.

Fig. R1. Cross-sectional SEM images of Zn deposited on Cu foil (a, c) with bilayer and (b, d) without SEI.

Our revision: According to the reviewer's comment, we have corrected the errors in the revised manuscript. The detailed revision is shown as follows: "*We performed a fast Fourier transform on each image (Fig. 2j-2l) and quantitatively evaluated their similarity to the cross-sections of every crystal in reciprocal space.*" and "*The high-resolution cross-sectional SEM image reveals that the bilayer SEI enables dense Zn deposition (actual thickness: 18.4 μm) close to the theoretical Zn deposition thickness of 17.1 μm under 10 mAh cm^{-2} (Fig. 5i) ⁷. However, Zn deposition layer without SEI results in highly porous, nodular Zn morphology (Fig. 5h and 5j), with a thickness of 16.3 μm , which is much lower than the theoretical thickness due to more severe side reactions during the Zn deposition process. This ultra-dense Zn deposition not only can reduce the contact area with the electrolyte, but also can maintain good electron transport and mass transport, thus reducing the generation of inactive Zn and improving the reversibility of Zn plating/stripping ⁴²*" Please see the highlighted part in yellow background on page 7, page 14 and Fig. 5 on page 15 in the revised manuscript.

Comment 2: Figure Correction

- 1) On page 13, the description for Fig. 4d and 4e should be corrected from "absorbed" to "adsorbed".
- 2) On page 15, it would be helpful to specify the observed values of SEI thickness on Fig. 5g and Fig. 5h.
- 3) On page 15, since SEI is commonly found in current batteries, it might be prudent to consider using 'without bilayer SEI' instead of 'without SEI' in Fig 5. Additionally, 'bare SEI' appears to be a suitable alternative term.
- 4) On page 17, As previously mentioned, it would be advisable to replace 'without SEI' in Fig. 6 with a more appropriate term such as 'bare SEI', or 'without bilayer SEI'.

Our response: Thanks a lot for the reviewer's careful reading and constructive comments. In accordance with the reviewer's comments, we have meticulously reviewed and corrected inaccuracies in descriptions throughout the manuscript.

- 1) Following the reviewer's comment, we have rectified the description of Fig. 4d and 4e on page 14 from "absorbed" to "adsorbed".

2) First, we would like to clarify that Fig. 5g and 5h depict cross-sectional SEM images of 10 mAh cm^{-2} Zn deposition on Cu foil under different electrolytes, where the labeled thickness represents the deposition thickness of Zn with/without the bilayer SEI, rather than the SEI thickness itself. Typically, the SEI thickness is only a few tens of nanometers and cannot be observed via SEM. In response to the reviewer's suggestions, we have annotated the thickness of Zn deposition with/without the bilayer SEI in Fig. 5g-5h (**Fig. R1**) and also supplemented explanations in the manuscript to facilitate observation of the differences in electrolyte conditions.

3) We would like to point out that there is commonly no formation of SEI on the Zn anode in the common Zn-based salt electrolyte system (such as ZnSO_4). The formation of SEI typically only occurs in modified electrolytes (such as those with electrolyte additives). This point has been widely reported and clearly explained in the literature (*Nat. Nanotechnol.* **16**, 902-910 (2021); *Joule* **6**, 1103-1120 (2022); *Angew. Chem. Int. Ed. Engl.* **61**, e202212839 (2022)). Furthermore, we also observed through TEM and XPS analysis that there is no observable formation of SEI on the Zn deposited in ZnSO_4 electrolyte (**Fig. R2**). Therefore, we believe it is appropriate to use “without SEI” to describe the Zn anode serving as a blank sample in the ZnSO_4 electrolyte in our manuscript.

Fig. R2. HRTEM images of Zn surface with no SEI formation in ZnSO_4 electrolyte.

Fig. R3. XPS spectra of Zn foil cycling in ZnSO₄ electrolyte. (a) C 1s and (b) S 2p after Ar⁺ sputtering from 0 s to 600 s.

4) Thanks for the reviewer’s valuable comment. As previously mentioned, we think “without SEI” is more appropriate than “bare SEI” or “without bilayer SEI” to describe the blank samples.

Our revision: According to the reviewer’s comment, we have corrected the errors in the revised manuscript. The detailed revision is shown as follows: *“As shown in Fig. 2a and Supplementary Fig. 8, no obvious interfacial layer can be found in Zn deposited in pristine ZnSO₄ electrolyte, because the decomposition products of solvated H₂O cannot form a dense layer to cover the surface of Zn anode^{15, 25, 26}.”* *“d-e, Differential charge density maps of (d) H₂O and (e) DMI adsorbed on ZnCO₃”* and *“The high-resolution cross-sectional SEM image reveals that the bilayer SEI enables dense Zn deposition (actual thickness: 18.4 μm) close to the theoretical Zn deposition thickness of 17.1 μm under 10 mAh cm⁻² (Fig. 5i)⁷. However, Zn deposition layer without SEI results in highly porous, nodular Zn morphology (Fig. 5h and 5j), with a thickness of 16.3 μm, which is much lower than the theoretical thickness due to more severe side reactions during the Zn deposition process.”* Please see the highlighted part in yellow background on page 7, page 13, page 14 and Fig.5 on page 15 in the revised manuscript.

Comment 3: More details

1) On page 9, the reference of the sentence “Pyridinic C-N species are formed through the electrochemical decomposition of DMI and pyrrolic C-N species are associated with DMI adsorption.” is required.

2) On page 9, the interpretation of the presence of SO_4^{2-} and SO_3^{2-} ions in the outer layer depicted in Fig. 3c appears to be insufficient. If their presence is attributed to the ZnSO_4 electrolyte, it would be beneficial to explicitly mention it for clarification.

3) On page 14, a reference is needed for the statement regarding the “theoretical Zn deposition thickness of $17.1 \mu\text{m}$ ”.

Our response: Thanks a lot for the reviewer’s constructive comment, which are beneficial to improving the quality of our manuscript. To further enhance the comprehensiveness of our manuscript, we have provided additional explanations and supplemented references in accordance with the reviewer's suggestions, which guides us toward improving the quality of our work. We would like to respond to the reviewer’s comments point-by-point in the following:

1) Following the reviewer's suggestion, we have supplemented reference (*Angew. Chem. Int. Ed. Engl.* **61**, e202212839 (2022)) to support the explanation that “Pyridinic C-N species are formed through the electrochemical decomposition of DMI and pyrrolic C-N species are associated with DMI adsorption” in the revised manuscript.

2) Thank you for your valuable comments. To investigate the source of S on the Zn foil surface after cycling, we also conducted XPS with Ar^+ sputtering analysis on zinc foil cycling in ZnSO_4 electrolyte. As shown in **Fig. R4a**, the S 2p spectrum of the Zn foil cycling in ZnSO_4 without SEI shows a weak signal of S before Ar^+ sputtering, showing no ZnS. Notably, the signal of SO_4^{2-} on the Zn foil without SEI is also very weak compared to the bilayer SEI sample. As the Ar^+ sputtering time increased, the signal of S for Zn foil without SEI rapidly diminished and almost disappeared. In stark contrast, the signals of S for the Zn foil with the bilayer SEI remained strong both before and after Ar^+ sputtering. Therefore, we can reasonably exclude the possibility that the numerous presence of SO_4^{2-} is due to residual ZnSO_4 on the surface.

Fig. R4. S 2p spectra of Zn foil cycling in different electrolytes after Ar^+ sputtering from 0 s to 600 s. (a) 2 M ZnSO_4 and (b) 2 M ZnSO_4 + DMI.

Regarding the consistently strong signals of SO_4^{2-} observed before and after Ar^+ sputtering on the Zn foil with bilayer SEI, we have combined experimental results and mechanistic hypotheses to conclude that this is due to the overall presence of organic sulfates in the bilayer SEI. Specifically, we observed that the signals of N also remained consistently strong both before and after sputtering, in addition to the S element signal (**Fig. R5**). This indicates that the bilayer SEI is overall rich in C/N organic materials. Further considering the formation pathway of the bilayer SEI (as shown in **Fig. R6**), it is likely that these organic materials are belong to organic sulfates.

Fig. R5. N 1s spectra of Zn foil cycling in 2 M ZnSO_4 + DMI electrolyte after Ar^+ sputtering from 0 s to 600 s.

Fig. R6. Possible bilayer SEI formation pathways.

Regarding the formation of SO_3^{2-} , we believe this is due to the reduction of SO_4^{2-} during the electrochemical process. From the S 2p spectrum, it can be observed that the signal of SO_3^{2-} appears alongside the signal of ZnS after Ar^+ sputtering. In electrochemistry, sulfate ions (SO_4^{2-}) can undergo reduction reactions. These reduction reactions typically involve the reduction of sulfate ions to sulfur compounds with fewer oxygen atoms (commonly, S^{2-}). On the electrode surface, sulfate ions accept electrons and undergo reduction reactions, leading to the generation of corresponding reduction products. The reduction reaction may involve multiple steps, and the specific reaction pathway depends on factors such as reaction conditions and electrode materials. The partial reduction of SO_4^{2-} to SO_3^{2-} in a mildly acidic environment has been reported in previous studies (*Nat Commun* **15**, 4303 (2024), **Fig. R7**).

Fig. R7. XPS of Zn anodes after 50 cycles in 1 m ZnSO₄ + 0.1 m HTFSI (Nat Commun **15**, 4303 (2024)).

3) We appreciate the reviewer's good suggestion. We agree with the reviewer's suggestion regarding the need for further explanation and supplemental references to support the statement about the “theoretical Zn deposition thickness of 17.1 μm ”. We calculated the theoretical deposition thickness based on the theoretical volumetric capacity of Zn and the areal capacity of Zn deposition we set in the experiment. In Fig. 5g-5j, we set the areal capacity of Zn deposition on the Cu foil substrate as 10 mAh cm⁻². Given that the theoretical volumetric capacity of Zn is 5855 mAh cm⁻³, thus, the theoretical Zn deposition thickness at 10 mAh cm⁻² is calculated as $10/5855 \times 10000 = 17.1 \mu\text{m}$. Additionally, in response to the reviewer's comment, we have supplemented the explanation of the calculation of the theoretical Zn deposition thickness in the manuscript by referencing relevant literature (*Angew. Chem. Int. Ed.* **62**, e202308454 (2023)).

Our revision: According to the reviewer's comment, we have provided additional explanations and added the literature (*Angew. Chem. Int. Ed.* **61**, e202212839 (2022); *Nat Commun* **15**, 4303 (2024); *Angew. Chem. Int. Ed.* **62**, e202308454 (2023)) to supplement the descriptions in the revised manuscript as follows: "Pyridinic C-N species are formed through the electrochemical decomposition of DMI and pyrrolic C-N species are associated with DMI adsorption¹⁴." "ZnS and trace amounts of SO₃²⁻ may be produced by the reduction of SO₄²⁻³¹." and "The high-resolution cross-sectional SEM image reveals that the bilayer SEI enables dense Zn deposition (actual thickness: 18.4 μm) close to the theoretical Zn deposition thickness of 17.1 μm under 10 mAh cm⁻² (Fig. 5i)⁷." Please see the highlighted part in yellow background on page 9 and page 14 in the revised manuscript.

Comment 4: Additional experiment

- 1) On page 6, fig. 2a was noted the absence of a distinct interfacial layer, but the explanation provided remains inadequate. Further investigation into the bare interfacial layer's structure via EDS mapping is imperative for a comprehensive understanding.
- 2) On page 12, Further analysis is required to contextualize the significance of the value (0.22 eV) mentioned in Fig. 4f. It would be advantageous to compare it with the migration energy barrier of the SEI layer in additive-free aqueous zinc battery for a more comprehensive understanding.
- 3) To demonstrate the effect of the unique bilayer structure in this experiment, it would be beneficial to include experiments comparing with Zn cells having single-layer SEI. When compared with Zn cells without SEI layer, the difference seems to focus more on the presence or absence of SEI rather than the effect of the bilayer structure.

Our response: Thanks for the reviewer's valuable comments. We would like to respond to the reviewer's comments point-by-point in the following:

- 1) Thanks a lot for the reviewer's comment. Transmission electron microscopy (TEM) is a very straightforward characterization method to determine whether an SEI is formed. In this work, the nanostructure of SEI was further determined via high-resolution transmission electron microscopy (HRTEM). As shown in **Fig. R2**, no

obvious interfacial layer can be found in Zn deposited in pristine ZnSO₄ electrolyte, because the decomposition products of solvated H₂O cannot form a dense layer to cover the surface of Zn anode. Therefore, we conclude that there is no SEI interfacial layer on the surface of Zn deposited in ZnSO₄ electrolyte, which is consistent with previously reported work. According to the reviewer's suggestions, we have supplemented our study with EDS mapping of the Zn deposited in the ZnSO₄ electrolyte using TEM. As shown in **Fig. R8**, the surface of the deposited Zn only exhibits significant signals for Zn and O elements, likely due to partial oxidation of the deposited Zn nanostructures in air. To further support our conclusion, we have also included XPS coupled with Ar⁺ sputtering. As illustrated in **Fig. R3**, the Zn foil cycled in ZnSO₄ electrolyte shows only a small amount of C and S elements before sputtering, and almost no C and S elements after sputtering. In summary, we conclude that there is no SEI on the Zn deposited in the ZnSO₄ electrolyte.

Fig. R8. **a**, STEM image of Zn deposition in ZnSO₄ electrolyte without SEI. **b-c**, The corresponding EDS mapping of **(a)**

2) Thank you for your valuable feedback. We would like to clarify that our study demonstrated through TEM and XPS that Zn deposited in additive-free ZnSO₄ electrolyte does not form an SEI. Therefore, we cannot provide additional information on the migration energy barrier of Zn²⁺ in the SEI layer for cells using additive-free Zn-based aqueous electrolyte.

3) Thank you for your thorough review and valuable suggestions. We would like to clarify that prior to this work, we designed an electrolyte additive capable of forming a single-layer SEI on the Zn anode surface. Although Zn anodes with a single-layer SEI

demonstrated superior cycling performance of 2000 cycles with an ACE of 99.84% compared to those without SEI (unpublished data), this electrolyte struggled to achieve high utilization of the Zn anode. For the convenience of the reviewer, we have included this unpublished data as **Fig. R9**.

Fig. R9. Characterization and electrochemical performance of Zn anode with single-layer SEI. **a**, HRTEM image of Zn deposition with single-layer SEI. **b**, The CE of Zn plating/stripping with single-layer SEI in Cu-Zn asymmetric cells at 1 mAh cm⁻² and 2 mA cm⁻². **c**, Long-term stability of high ZUR symmetric cells at 28.4 mAh cm⁻² and 2 mA cm⁻². (Unpublished data)

Compared to single-layer SEI, we designed a robust bilayer SEI that simultaneously achieves homogeneous Zn²⁺ transport and excellent mechanical stability for record ZUR and ACE of Zn anode (**Fig. R10**). This bilayer SEI on Zn surface consists of a crystalline ZnCO₃-rich outer layer and an amorphous ZnS-rich inner layer. The ordered outer layer improves the mechanical stability during cycling, and the amorphous inner layer homogenizes Zn²⁺ transport for homogeneous, dense Zn deposition. As a result, the bilayer SEI enables highly reversible Zn plating/stripping with a record average CE of 99.95%. Impressively, Zn-Zn symmetric cells show excellent lifetime for over 550 h with a high ZUR of 98% under an unprecedented areal capacity of 28.4 mAh cm⁻², which are much superior to the values reported in the

literature. Based on the above comparisons (**Table R1**), we believe that our designed double-layer SEI structure plays a crucial role in achieving ultra-high CE and ZUR for the Zn anode.

Fig. R10. Electrochemical performance of Zn anode with bilayer SEI. **a**, The CE of Zn plating/stripping in Cu-Zn asymmetric cells at 1 mAh cm^{-2} and 2 mA cm^{-2} . **b**, Long-term stability of high ZUR symmetric cells at 28.4 mAh cm^{-2} and 2 mA cm^{-2} . **c**, Comparison of ZUR and lifetime between our work and previous studies on high ZUR.

Table R1. Summary of the electrochemical performance for different Zn anodes.

SEI	ACE	ZUR	Areal capacity	Lifetime
This work	99.95%	98%	28.4 mAh cm^{-2}	550 h
Single SEI	99.84%	98%	28.4 mAh cm^{-2}	70 h

Our revision: According to the reviewer’s comment, we have provided additional explanations in Supplementary Fig. 8 and Supplementary Fig. 18 to supplement the descriptions in the revised manuscript as follows: “*As shown in Fig. 2a and Supplementary Fig. 8, no obvious interfacial layer can be found in Zn deposited in pristine ZnSO_4 electrolyte, because the decomposition products of solvated H_2O cannot form a dense layer to cover the surface of Zn anode* ^{15, 25, 26}” and “*In contrast, the Zn*

foil cycled in ZnSO₄ electrolyte shows almost no signals of C and S elements after sputtering, further confirming that no SEI is formed in ZnSO₄ electrolyte (Supplementary Fig. 18)". Please see the highlighted part in yellow background on page 6, page 9 in the revised manuscript and Supplementary Fig. 8 on page 9, Supplementary Fig. 18 on page 19 in the revised Supplementary Information.

Comment 5: Supporting information

1) Adding a brief explanation in Supplementary Fig. 2 elucidating the reason behind the performance decline with increasing DMI concentration would be beneficial.

Our response: We appreciate the reviewer's insightful comment. The decline in the electrochemical performance of the Zn anode with increasing DMI concentration may be due to the following reasons:

1. The formation of a double-layer SEI is driven by the decomposition of DMI, which is consumed in the process. Therefore, a high concentration of the DMI additive can lead to the formation of an excessively thick SEI layer, increasing the difficulty for Zn²⁺ ions to pass through the SEI layer and thereby increasing the polarization of the battery (**Fig. R11**).

2. High concentrations of the DMI as additive may cause the SEI layer to form unevenly on the electrode surface, creating localized Zn²⁺ ions concentration gradients. This can result in non-uniform Zn deposition, increasing the risk of dendrite formation.

3. At high concentrations, DMI can reduce the overall conductivity of the electrolyte, affecting the ion transport efficiency within the electrolyte and thereby reducing the electrochemical performance of the battery (**Fig. R12**).

Fig. R11. Long-term stability of symmetric cells in 2 M ZnSO_4 electrolytes with different concentrations of DMI at 5 mA cm^{-2} and 10 mA cm^{-2} .

Fig. R12. The conductivity of 2 M ZnSO_4 electrolytes with different DMI concentrations.

Our revision: Based on the reviewer's comments, we have added a brief explanation in Supplementary Fig. 2 and Supplementary Fig. 3 to elucidate the reasons for the decline in performance with increasing DMI concentration. The detailed revision is shown as follows: “*With the increase in DMI concentration, the cycling life of the battery significantly decreases, and the overpotential significantly increases. This may be due to (1) the decrease in the ionic conductivity of the electrolyte (as shown in Supplementary Fig. 3) and (2) the excessively thick SEI formed by the high concentration of DMI decomposition, leading to difficult and uneven Zn^{2+} transport.*” and “*Supplementary Fig. 3. The conductivity of 2 M ZnSO_4 electrolytes with different DMI concentrations*” Please see the highlighted part in yellow background on page 3 and Supplementary Fig. 3 on page 4 in the revised Supplementary Information.

Reviewer #2 (Remarks to the Author):

General Comment: In this manuscript, the authors use 1,3-dimethyl-2-imidazolidinone (DMI) as an electrolyte additive for aqueous zinc-ion batteries. However, previous research has reported the use of DMI as an electrolyte additive for aqueous zinc-ion batteries (DOI: 10.1016/j.cej.2023.141287). Even this study provides new insights into the SEI formed by DMI, it is lack of novelty. Here, DMI induced a robust amorphous/crystalline bilayer SEI that achieves the desirable homogeneous Zn^{2+} transport and mechanical stability to achieve record ZUR and CE. The full cells based on the bilayer SEI anode show better cycling performance under low N/P ration. Several crucial issues need further exploration by the authors before next submission.

Our response: We gratefully appreciate the reviewer's careful reading and positive comments to our manuscript, which are beneficial to improving the quality of our manuscript. Firstly, we would like to emphasize the novelty and innovation of our work in constructing a bilayer SEI on the Zn anode to achieve record CE and Zn anode utilization. While DMI as an additive has been reported before, our work is entirely different from previous studies.

Earlier studies on DMI as an additive primarily highlighted that the inclusion of DMI led to the formation of a new Zn^{2+} solvation structure, disrupting the original hydrogen bond network in the system and suppressing side reactions caused by the Grotthuss mechanism. Although these previous works suggested that DMI as additive could form an SEI layer on the Zn anode surface, they did not explain the structure, composition, or formation mechanism of the SEI. In stark contrast, our work proposes that the addition of an appropriate amount of DMI forms a unique bilayer SEI on the surface of the Zn anode. We thoroughly investigated the structure, composition, and formation mechanism of this bilayer SEI using a series of novel and detailed characterizations. The details are listed as follows:

Novelty of the bilayer SEI structure: This bilayer SEI on Zn surface consists of a crystalline $ZnCO_3$ -rich outer layer and an amorphous ZnS-rich inner layer. The ordered outer layer improves the mechanical stability during cycling, and the

amorphous inner layer homogenizes Zn^{2+} transport for homogeneous, dense Zn deposition. This structure of SEI for Zn anode has never been reported before.

Comprehensive and innovative characterizations: We employed a series of detailed and novel characterizations to explore the structure, composition, and formation mechanism of the bilayer SEI. It is particularly noteworthy that this work innovatively utilizes big data methods for an in-depth analysis of HRTEM data. Due to the multifarious possibilities of SEI components and the complexity of HRTEM images, it is not reliable enough to ascribe a crystal structure from an image by simply measuring its lattice parameters and matching with the existing ones. Thorough structural identification requires the consideration of all possible projections of reasonable structures and careful comparison of lattice information from images. Therefore, we selected three representative regions (referred to as Areas 1-3) from the HRTEM images and conducted a thorough structural comparison with 187 related crystal structures in the Materials Project and ICSD databases using statistical analyses. We performed a fast Fourier transform on each image and quantitatively evaluated their similarity to the cross-sections of every crystal in reciprocal space. Statistical analyses show that the images of the crystalline region of SEI closely match with three structures among the studied 187 ones. Through these extensive and reliable methods, we can conduct a detailed analysis of the SEI composition.

Record utilization and CE of Zn anode. We have designed a robust bilayer SEI that simultaneously achieved the homogeneous Zn^{2+} transport and mechanical stability for record Zn utilization rate (ZUR) and CE of Zn anode. The synergistic bilayer SEI enables a record-high average CE (99.95%) of Zn plating/stripping. Impressively, Zn-Zn symmetric cells can achieve stable cycling for more than 550 h with an extremely high ZUR of 98% under a large areal capacity of 28.4 mAh cm^{-2} , which is much superior to the values reported in the literature.

Therefore, our manuscript is of high novelty, which is fundamentally and completely different from all the previously reported work.

Moreover, we have conducted additional experiments as per the reviewer's good comments and made considerable revisions to improve the clarity and readability of the

manuscript. All changes have been highlighted in yellow background in the revised manuscript and supporting information. We are confident that these changes have strengthened the quality of our work and we hope that the revised manuscript meets the standards required for publication in *Nature Communications*.

Comment 1: Why does the concentration of DMI have such a significant impact on battery performance? As depicted in Supplementary Figure 2, the performance of cells using 50 mM and 100 mM DMI is considerably worse than those using 10 mM. Consequently, exploring concentrations lower than 10 mM may also be worthwhile.

Our response: Thanks a lot for the reviewer's careful reading and valuable comment. The performance of cells using 50 mM and 100 mM DMI is considerably worse than those using 10 mM, which is due to the following reasons: 1. The formation of a double-layer SEI is driven by the decomposition of DMI, which is consumed in the process. Therefore, a high concentration of the DMI additive can lead to the formation of an excessively thick SEI layer, increasing the difficulty for Zn^{2+} ions to pass through the SEI layer and thereby increasing the internal resistance of the battery. 2. High concentrations of the DMI as additive may cause the SEI layer to form unevenly on the electrode surface, creating localized Zn^{2+} ions concentration gradients. This can result in non-uniform Zn deposition, increasing the risk of dendrite formation. 3. At high concentrations, DMI can reduce the overall conductivity of the electrolyte, affecting the ion transport efficiency within the electrolyte and thereby reducing the electrochemical performance of the battery (**Fig. R1**). Based on the above reasons, the electrochemical performance of the Zn-Zn symmetric cell significantly decreases when the DMI concentration is 50 mM and 100 mM.

Fig. R1. The conductivity of 2 M ZnSO₄ electrolytes with different DMI concentrations.

Furthermore, we have supplemented extra electrochemical tests to explore the electrochemical performance of electrolytes with lower DMI concentrations. In the Zn-Zn symmetric cells, the overpotentials in electrolytes with 1 mM and 5 mM DMI are lower than that with 10 mM DMI, but their cycling performance is significantly lower than that with 10 mM DMI (**Fig. R2**). Furthermore, the CE and cycling performance of Cu-Zn asymmetric cells with low concentration DMI electrolytes are markedly lower than those with the optimal concentration of 10 mM (**Fig. R3**). Therefore, based on the data from both symmetric and asymmetric cells, we have determined that the optimal concentration of DMI is 10 mM (**Table. R1**).

Fig. R2. Long-term stability of Zn-Zn symmetric cells in 2 M ZnSO₄ electrolytes with low concentrations of DMI at 5 mAh cm⁻² and 10 mA cm⁻².

Fig. R3. The CE of Zn plating/stripping in Cu-Zn asymmetric cells with different DMI concentration at 1 mAh cm⁻² and 2 mA cm⁻².

Table R1. Summary of the electrochemical performance for Zn anodes with different low DMI concentrations.

DMI concentration (mM)	Lifetime of Zn-Zn cell (h)	Lifetime of Cu-Zn cell (cycles)	ACE
1	100	693	99.67%
5	325	925	99.73%
10	1600	4500	99.95%

Our revision: Based on the reviewer's comments, we have added electrochemical tests for low concentrations of DMI and have revised and supplemented the explanation for Supplementary Fig. 2. The detailed revision is shown as follows: “*Considering all factors, an optimal concentration of DMI was determined to be 10 mM in the ZnSO₄ electrolyte (Supplementary Figs. 2-3)*”, “*With the increase in DMI concentration, the cycling life of the battery significantly decreases, and the overpotential significantly increases. This may be due to (1) the decrease in the ionic conductivity of the electrolyte (as shown in Supplementary Fig. 3) and (2) the excessively thick SEI formed by the high concentration of DMI decomposition, leading to difficult and uneven Zn²⁺ transport.*” and “*Supplementary Fig. 3. The conductivity of 2 M ZnSO₄ electrolytes with different*

DMI concentrations” Please see the highlighted part in yellow background on page 6 in the revised manuscript, page 3 and Supplementary Fig. 3 on page 4 in the revised Supplementary Information.

Comment 2: Is the CO₂ produced by the decomposition of 10 mM DMI sufficient to form a uniform ZnCO₃ layer? The solubility of CO₂ is known to be around 1.49 g L⁻¹ in low concentration aqueous electrolytes under ambient temperature and pressure, similar to that in water. However, ZnCO₃ is not typically observed on the surface of the zinc metal anode in conventional electrolytes. The author should give further explanation on this problem, it is better to provide some calculation formulas to prove it.

Our response: Thanks a lot for the reviewer’s critical comment. Based on our experimental findings, we would like to point out that the decomposition of 10 mM DMI is sufficient to generate a uniform outer layer of ZnCO₃ in the SEI. We have proposed in Fig. 4b the possible mechanisms of the formation of various components in the designed bilayer SEI and the overall equation involving DMI in the reaction. For the reviewer's convenience, we have included Fig. 4b as **Fig. R4** here. Despite the relatively low concentration of added DMI at 10 mM, ZnCO₃ is predominantly situated in the outer layer of the bilayer SEI, with a thin thickness of approximately 10-15 nm, as depicted in **Fig. R5**.

Fig. R4. Possible bilayer SEI formation pathways.

Fig. R5. a-b, HRTEM images of bilayer SEI on the Zn anode.

Secondly, we would like to point out that under ambient temperature and pressure, the solubility of CO₂ in low-concentration aqueous electrolytes is approximately 1.49 g L⁻¹, similar to that in water. This result is based on the assumption of complete saturation of pure CO₂ in water. However, the actual concentration of CO₂ in air is approximately 0.03%-0.04%, resulting in a typical CO₂ concentration in natural water of less than 0.01 g L⁻¹. Furthermore, we have computed that assuming complete decomposition of 10 mM/L DMI results in a CO₂ concentration of approximately $10 \times 44 / 1000 = 0.44$ g L⁻¹, significantly higher than that of conventional Zn-based electrolytes.

Moreover, we argue that the concentration of CO₂ in the solution is not the sole factor influencing the formation of ZnCO₃. If CO₂ present in the solution was the primary source, its diffusion to the surface of Zn anode, adsorption, and subsequent combination with Zn²⁺ to form ZnCO₃ would be a lengthy process, typically not observed in conventional ZnSO₄ electrolytes (*Adv. Mater.* 2023, 2210051). However, in our designed electrolyte, the stronger adsorption between DMI and the Zn anode facilitates direct DMI adsorption on the Zn anode and subsequent decomposition (**Fig. R6**). This effectively circumvents the prolonged solution diffusion process, thereby favoring the formation of ZnCO₃-containing SEI.

Fig. R6. The adsorption energy of H₂O/DMI on Zn and ZnCO₃ surfaces.

Comment 3: Regarding ZnS in the SEI layer, the authors attribute it to the further reaction of S²⁻, which is the product after the reduction of adsorbed SO₄²⁻. However, the authors do not provide a detailed explanation. I recommend that the authors find sufficient supporting evidence, such as references, or demonstrate the plausibility of this reaction through Density Functional Theory (DFT) calculations. In addition, SO₄²⁻ is also adsorbed on the surface of the zinc metal anode in conventional electrolyte. Why is ZnS not observed in conventional electrolyte?

Our response: Thanks a lot for the reviewer's critical comment and good suggestions. Regarding our conclusion that the ZnS in the SEI layer is a product of the reduction of SO₄²⁻ adsorbed on the Zn anode surface, we have provided additional explanations based on the reviewer's suggestions and cited relevant references.

In electrochemistry, SO₄²⁻ ions undergo reduction reactions, typically being reduced to sulfur compounds with fewer oxygen atoms (commonly S²⁻). This phenomenon has been reported in industrial wastewater treatment (*AIChE J.* **67**, e17309 (2021)). The conclusion that SO₄²⁻ in ZnSO₄ + additive electrolytes can be reduced to form ZnS-containing SEI was first demonstrated by Nazar et al. in ZnSO₄ + DOTf electrolytes, where SO₄²⁻ was reduced to ZnS in a localized acidic environment (**Fig. R7a**, *Joule* **6**, 1733-1738 (2022)). This viewpoint has been corroborated by subsequent studies on SEI formation on Zn anodes (*ACS Nano* **17**, 552-560 (2022); *Journal of Energy Chemistry* **79**, 450-458 (2023)).

Fig. R7. High-resolution S 2p XPS spectra with Ar⁺ sputtering of Zn anode cycled in different electrolytes: **(a)** ZnSO₄ + DOTf (Joule **6**, 1733-1738 (2022)), **(b)** ZnSO₄ + cysteine (*Journal of Energy Chemistry* **79**, 450-458 (2023)) and **(c)** ZnSO₄ + Gly (*ACS Nano* **17**, 552-560 (2022)).

It is important to note that this reaction requires the presence of protons (H⁺). The continuous decomposition of DMI may lead to the formation of a strongly acidic local environment at the electrode-electrolyte interface, thereby promoting the reduction of SO₄²⁻ to form ZnS. In pure ZnSO₄ electrolytes, however, the intense HER tends to create a locally alkaline environment on the Zn anode surface, resulting in the formation of basic zinc sulfate and other byproducts. Therefore, ZnS is not observed in traditional ZnSO₄ electrolytes.

Our revision: According to the reviewer’s comment, we have added the literature (*AIChE J.* **67**, e17309 (2021); *Journal of Energy Chemistry* **79**, 450-458 (2023); *ACS Nano* **17**, 552-560 (2022)) as Ref. [34], Ref. [35] and Ref. [36] in the revised manuscript as following: “*ZnS and trace amounts of SO₃²⁻ may be produced by the reduction of SO₄²⁻ 31*” And “*In the meanwhile, a small portion of SO₄²⁻ adsorbed on the surface of the Zn anode undergoes reduction to S²⁻ in the electron-driven and proton-driven processes 34, 35. The S²⁻ further combines with Zn²⁺ to form ZnS 36*” Please see the

highlighted part in yellow background on page 9, page 12 and page 26 in the revised manuscript.

Comment 4: Unlike Supplementary Figure 28, Supplementary Figure 29 shows a significant presence of Cu elements distributed in the surface layer of the sample. I recommend that the authors re-test to verify this issue.

Our response: Thank you for the reviewer's thorough reading and valuable comments. Regarding the issue of the large amount of copper elements distributed on the sample surface (**Fig. R8**), we believe it may be due to the following reasons:

1. To avoid stress-induced damage to the cross-section from cutting the sample with scissors or a blade, we used an ion beam polish system instrument to obtain smooth cross-section samples. The principle involves bombarding the sample surface with a high-energy ion beam. When the high-energy ions elastically collide with the surface atoms of the sample, and the energy of these surface atoms increases beyond their work function, the atoms are ejected from the sample, causing the surface to gradually lose atoms and become thinner. During this process, Zn and Cu atoms inevitably undergo free sputtering, leading to the detection of Cu elements in the Zn region and Zn elements in the Cu region. Additionally, before placing the sample into the ion beam thinning instrument, it needs to be fixed and protected with Cu tape (**Fig. R9**), which increases the amount of Cu atom sputtering during the thinning process. To mitigate the impact of this sample preparation method, we re-obtained new cross-section samples by directly cutting them with scissors. As shown in **Fig. R10**, although the cross-section is not very smooth and there are height differences, the intensity of the Cu element signal on the sample surface has significantly weakened.

Fig. R8. **a**, Cross-sectional SEM image of Zn deposited on Cu foil with bilayer SEI at 10 mAh cm⁻². **b-d**, EDS mapping of **(a)**. This cross-section sample is obtained by using an ion beam polish system.

Fig. R9. Digital photos of conductive Cu tape.

Fig. R10. **a**, Cross-sectional SEM image of Zn deposited on Cu foil with bilayer SEI at 10 mAh cm⁻². **b-d**, EDS mapping of **(a)**. This cross-section sample is obtained by using a scissor.

2. Element overlap: The X-ray energy peaks of Cu and Zn are quite close, which can lead to overlap. The K_α peak of Cu is at 8.04 keV, while the K_α peak of Zn is at 8.64 keV (**Fig. R11**). This proximity in energy peaks can make it difficult to distinguish between these two elements in EDS analysis. This may also explain why, even when using scissors for sample preparation, Cu elements are still detected on the sample surface (although reduced, they are still present). To further confirm this, we directly deposited 10 mAh cm² of Zn onto Cu foil and performed EDS mapping on the surface (Sample 1, Sample 2, Sample 3). Impressively, when we selected both potential elements (Cu and Zn), a strong Cu signal was still observed on the sample surface (**Fig. R12-Fig. R14**). At this point, elemental content analysis showed that the Cu element content was 0, and the Zn element content was 100%.

Fig. R11. EDS spectrum of Zn deposited on Cu foil corresponding to Fig. R9.

Therefore, in summary, we believe that the appearance of Cu elements on the surface during cross-sectional testing of Zn deposition on Cu foil is quite normal. We want to emphasize that this phenomenon is frequently observed in our work. The absence of elemental distribution on the surface in the EDS mapping of the cross-section without SEI is due to the significant height differences, which cause the EDS detector to pick up signals that are very weak, almost negligible (**Fig. R15**).

Fig. R12. **a**, Top view SEM image of Zn deposited on Cu foil with bilayer SEI at 10 mAh cm^{-2} . **b-d**, EDS mapping of **(a)**. (Sample 1)

Fig. R13. **a**, Top view SEM image of Zn deposited on Cu foil with bilayer SEI at 10 mAh cm^{-2} .
b-d, EDS mapping of **(a)**. (Sample 2)

Fig. R14. **a**, Top view SEM image of Zn deposited on Cu foil with bilayer SEI at 10 mAh cm^{-2} .
b-d, EDS mapping of **(a)**. (Sample 3)

Fig. R15. **a**, Cross-sectional SEM image of Zn deposited on Cu foil without SEI at 10 mAh cm^{-2} . **b-d**, EDS mapping of **(a)**.

Comment 5: There appear to be some anomalies in Fig. 5l, notably a clear horizontal line in the middle of the image, which may be caused by a sudden change in the height of the sample. I suggest that the authors change the X-axis scanning direction of AFM and re-test the sample.

Our response: Thanks a lot for the reviewer's careful reading and constructive comment. We agree with the reviewer's suggestion to retest the AFM in Fig. 5l. Previously, the AFM images in Fig. 5k and 5l were based on Zn deposition on Cu foil, which may have led to signal disturbances during testing due to potential imperfections in the smoothness of the Cu foil itself. To avoid this issue, we have retested the AFM on Zn deposited on Cu-sputtered ITO, as shown in **Fig. R16**. New AFM images in Fig. 5k and 5l exhibits no sudden signal disturbances in the revised manuscript.

Fig. R16. AFM topography images of Zn deposited on Cu foil (a) with bilayer and (b) without SEI.

Our revision: According to the reviewer's comment, we have retest the AFM test of Fig. 5k and 5l t in the revised manuscript as follows: “*AFM topography images of Zn deposited on Cu foil (a) with bilayer and (b) without SEI*” Please see the highlighted part in yellow background on page 15 in the revised manuscript.

Comment 6: Based on Fig. 2d-I, it is challenging to conclude that the bilayer SEI is composed of the crystalline ZnCO_3 -rich outer layer and amorphous ZnS -rich inner layer, while its overall components are dominated by C/N-rich organics. Could the authors provide more images to support this conclusion?

Our response: We appreciate the reviewer's comment. We would like to clarify that our conclusion regarding "the bilayer SEI is composed of the crystalline ZnCO_3 -rich outer layer and amorphous ZnS -rich inner layer, while its overall components are dominated by C/N-rich organics" was not solely based on the TEM results shown in Fig. 2. We derived insights into the nanostructure, elemental composition, and main components of the bilayer SEI through TEM analysis. As depicted in **Fig. R17**, based on HRTEM analysis, we observed that the addition of DMI led to the formation of a bilayer SEI on the surface of the Zn anode, which consists of an outer crystalline layer

and an inner amorphous layer. EDS mapping further revealed that the bilayer SEI contains five elements: Zn, C, N, O, and S. By matching the fast Fourier transform patterns of the outer crystalline layer with the reciprocal space of substances possibly composed of these five elements, we concluded that the crystalline outer layer of the bilayer SEI is defined to be ZnCO₃-rich.

Fig. R17. Characterization of nanostructure and composition of SEI. **a**, HRTEM images of Zn surface with no SEI formation in ZnSO₄ electrolyte. **b-c**, HRTEM images of bilayer SEI on the Zn anode. **d**, STEM image of bilayer SEI on the Zn anode. **e-i**, The corresponding EDS mapping of (**d**). **j-l**, Fast Fourier transform on three representative regions of the SEI outer layer. **m-u**, Actual vs. theoretical reciprocal space of (**m-o**) ZnO-mp-2133, (**p-r**) ZnCO₃-mp-9812 and (**s-u**) Zn(CO₂)₂-mp-559437 in different crystal regions.

Regarding the chemical composition of the bilayer SEI, we obtained information through XPS coupled with Ar⁺ sputtering and ToF-SIMS analysis, as illustrated in Fig. 3. **Fig. R18** shows the XPS spectra of C 1s, N 1s and S 2p after Ar⁺ sputtering from 0 s to 600 s. The peak at 288.6 eV is attributed to the ZnCO₃, while the peaks at 285.4 and 284.7 eV are attributed to the C-O/C-N and C-C/C-H bonds, respectively. We attribute these organic species to the electrochemical decomposition/surface adsorption of DMI. As the Ar⁺ sputtering consistently proceeds, the signals of C-C/C-H and C-O/C-N species drop sharply, and the signals of ZnCO₃ species also become weaker in the spectra. This indicates that the outer layer of SEI is comprised of ZnCO₃, which is consistent with the TEM analysis of crystalline outer layer. From the N 1s spectrum, it is evident that before Ar⁺ sputtering, prominent signals appeared at 400.4 eV for pyrrolic C-N and at 398.8 eV for pyridinic C-N. Impressively, as the sputtering process progresses, the content of N does not decrease but slightly increases (**Fig. R19**), indicating the formation of a uniform, dense C/N organic layer rich in nitrogen throughout the entire SEI layer. Furthermore, the S 2p spectrum before Ar⁺ sputtering confirms no presence of ZnS, which represents the outer layer of SEI is essentially free of ZnS. Impressively, the amount of ZnS increases significantly with the sputtering depth, suggesting that the inner layer of SEI is ZnS-rich. Based on these XPS results, we summarize that the bilayer SEI is composed of the crystalline ZnCO₃-rich outer layer and amorphous ZnS-rich inner layer, while its overall components are dominated by C/N-rich organics.

Fig. R18. Interfacial studies of chemical composition. a-c, XPS spectra of (a) C 1s, (b) N 1s, and (c) S 2p after Ar⁺ sputtering from 0 s to 600 s.

Fig. R19. Atomic concentrations of C, N, and S during the XPS Ar⁺ sputtering process.

Furthermore, the three-dimensional (3D) spatial distribution of organic component of the bilayer SEI was further investigated by the depth profile analysis of ToF-SIMS (Fig. R20). CH⁻ and CN⁻ are characteristic ionic fragments of organic components. With the sputtering time increasing, the content of CH⁻ drops rapidly and almost disappears, which is concordant with C 1s spectrum. In addition, the CN⁻ organic

components in SEI distribute uniformly along the thickness and are likely compact to form a robust layer, which is in good buffering and connectivity for the entire SEI.

Fig. R20. 3D views of organic components in the ToF-SIMS of bilayer SEI.

Thus, combining the TEM, XPS and ToF-SIMS results, we conclude that in the DMI additive electrolyte, Zn anode can form a unique bilayer SEI structure with the ZnCO₃-rich outer layer, ZnS-rich inner layer and overall C/N-rich organic bulk phase.

Comment 7: On page 16, the author states, "...Linear sweep voltammetry (LSV) curves of the Zn anode with/without bilayer SEI were measured (Supplementary Fig. 30). The result indicates that the HER is significantly suppressed on the bilayer SEI of Zn anode." According to the reaction equation for SEI formation: $C_5H_{10}N_2O + 2ZnSO_4 + 8H^+ + 8e^- \rightarrow ZnCO_3 + ZnS + [C_4H_{14}N_2]SO_4 + H_2O$, both ZnCO₃ and ZnS come from ZnSO₄. However, in Supplementary Fig. 30, the LSV curves of Zn were obtained from 2 M Na₂SO₄ and 2 M Na₂SO₄ + 10 mM DMI, respectively. That is to say, there is no ZnSO₄ in these electrolytes, so what is the composition of the SEI? It shouldn't be NaCO₃ or Na₂S because they can dissolve in water.

Our response: Thanks a lot for the reviewer's careful reading and valuable comments. Regarding the explanation of the HER testing, we apologize for the incorrect Supplementary Fig. 30 and experimental section. In fact, for the HER testing, we used 2 M Na₂SO₄ as the electrolyte for both the blank and experimental samples. The only difference between the two samples lies in the working electrode used for HER test. Specifically, the blank samples employed a bare Zn foil as the working electrode, while the experimental sample used a Zn foil that had undergone several cycles in 2 M ZnSO₄

+ 10 mM DMI electrolyte, forming a bilayer SEI on the surface of the Zn foil. Upon reflecting on the reviewer's comments, we acknowledge the error in the description in the manuscript and have accordingly corrected the caption of Supplementary Fig. 30 and the corresponding section in the experimental part. For the reviewer's convenience, we have included revised Supplementary Fig. 30 as **Fig. R21** here.

Fig. R21. LSV curves of bare Zn or Zn foil with bilayer SEI in 2 M Na₂SO₄ electrolyte at 5 mV s⁻¹.

Our revision: According to the reviewer's comment, we have rectified the description of the HER test in the experimental section and Supplementary Fig. 34 in revised manuscript and supplementary information. The detailed revision is shown as follows: *“For HER test, the bare Zn foil and Zn foil with bilayer SEI (after cycling in 2 M ZnSO₄ + 10 mM DMI electrolyte) were separately used as the working electrode, Pt as counter electrode and Ag/AgCl as reference electrode in 2 M Na₂SO₄ electrolyte.”* and *“Supplementary Fig. 34. LSV curves of bare Zn or Zn foil with bilayer SEI in 2 M Na₂SO₄ electrolyte at 5 mV s⁻¹.”* Please see the highlighted part in yellow background on page 23 in the revised manuscript, Supplementary Fig. 34 on page 35 in the revised supplementary information.

Comment 8: In this manuscript, the I₂ cathode were prepared by mixed I₂ and AC according to a mass ration of 1:4. Since AC has a high content, the authors should demonstrate the contribution of AC to specific capacity of the cathode.

Our response: Thanks a lot for the reviewer's critical comment. To verify the contribution of AC in the I₂@AC cathode, we assembled Zn-I₂ full cells and Zn-AC full cells with the same mass of AC (1.8 mg) and tested them under the same rate and current density. As shown in **Fig. R22**, under identical test conditions, the capacity of the Zn-AC full cell is 0.04605 mAh cm⁻², while the capacity of the Zn-I₂ full cell is 0.1261 mAh cm⁻². Therefore, the contribution of AC to the capacity in the I₂@AC cathode is 0.04605/0.1261*100% = 36.5%. After deducting the contribution of AC, the specific capacity of I₂ in the I₂@AC cathode is calculated as (0.1261-0.04605)/0.6*1000 = 133.4 mAh g⁻¹.

Fig. R22. GCD curves of Zn-I₂ and Zn-AC full cell with the same mass loading of AC at the same current density (The rate of 10 C is based on the mass of I₂).

Comment 9: The authors should provide more experiment details, such as the TEM sample preparation process, the electrolytes of Zn-Br₂ full cells, and calculation method of N/P ratio.

Our response: Thanks for the reviewer's valuable comment. We agree with the reviewer's suggestion that we should provide more experimental details, such as the TEM sample preparation process, the electrolytes of Zn-Br₂ full cells, and calculation method of N/P ratio.

Regarding the preparation method of TEM samples, we employed an in-situ electro-deposition approach, as depicted in **Fig. R23**. A Cu TEM grid was clamped with a reverse clamp and immersed in the electrolyte as the working electrode, with Zn

foil serving as the counter electrode and reference electrode. Zn was then electro-deposited in-situ onto the Cu grid. After deposition, the Cu grid was immediately rinsed with water and ethanol.

Fig. R23. Digital photographs of the in-situ TEM setup for Zn electrodeposition.

As for the electrolyte of anode-free Zn-Br full cell, we used 2 M ZnSO_4 + 10 mM DMI + 0.5 M ZnBr_2 .

It is widely known that the N/P ratio refers to the ratio of the capacity of anode to that of cathode of battery. Regarding the calculation method of the N/P ratio in this work, we computed the theoretical capacity of the cathode. Then, a predetermined amount of Zn was pre-deposited on the Cu foil for the anode to achieve the set N/P ratio. Specifically, we weighed the mass of the active material in the cathode and calculated the theoretical capacity based on its theoretical specific capacity. Assuming an N/P ratio of 2, the capacity of Zn pre-deposited on the Cu foil for the cathode would be twice that of the corresponding cathode.

Our revision: According to the reviewer's comment, we made changes to the revised manuscript as shown in the following. In the revised manuscript, we have added the details as follows: *“Regarding the preparation method of TEM samples, we employed an in-situ electro-deposition approach. A Cu TEM grid was clamped with a reverse clamp and immersed in the electrolyte as the working electrode, with Zn foil serving as the counter electrode and reference electrode. Zn was then electro-deposited in-situ onto the Cu grid. After deposition, the Cu grid was immediately rinsed with water and*

ethanol.”, “*Anode-free Zn-Br full cells were assembled by Cu foil as anode and Br cathode with 2 M ZnSO₄ + 10 mM DMI + 0.5 M ZnBr₂ electrolyte*”, “*Regarding the calculation method of the N/P ratio in this work, we computed the theoretical capacity of the cathode. Then, a predetermined amount of Zn was pre-deposited on the Cu foil for the anode to achieve the set N/P ratio. Specifically, we weighed the mass of the active material in the cathode and calculated the theoretical capacity based on its theoretical specific capacity. Assuming an N/P ratio of 2, the capacity of Zn pre-deposited on the Cu foil for the cathode would be twice that of the corresponding cathode.*” and “**Supplementary Fig. 7. Digital photographs of the in-situ TEM setup for Zn electrodeposition.**”. Please see the highlighted part in yellow background on page 21, page 22 in the revised manuscript and Supplementary Fig. 7 on page 8 in the revised Supplementary Information.

Reviewer #3 (Remarks to the Author):

General Comment: Meng and coworkers reported a robust bilayer SEI for protecting zinc anode, which features homogeneous Zn^{2+} transport, excellent mechanical rigidity, record Zn utilization rate, and high coulombic efficiency. Moreover, the assembled Zn-Zn symmetric cells show excellent lifetime with a high ZUR and the Zn full cell exhibit superior electrochemical stability at a low N/P ratio. Different from conventional single-layer SEIs that suffer from easily rapture and failure, this zinc anode with unique bilayer SEI with bilayer design provides a promising alternative solution for addressing the key challenges faced by the anode of aqueous zinc batteries. This work is interesting and well-organized, and the given characterization results supports their hypothesis. Although this work has many interesting results, I still have some following concerns.

Our response: We gratefully appreciate the reviewer's careful reading and positive comments to our manuscript. The manuscript has been thoroughly revised according to the reviewer's insightful suggestions as detailed below. All changes have been highlighted in yellow background in the revised manuscript and supporting information. We hope that the revised manuscript is suitable for publication in *Nature Communications*.

Comment 1: As there are so many reported works about the in-situ forming SEI for zinc anode by introducing electrolyte additives, what are the advantages of your 1,3-Dimethyl-2-imidazolidinone (DMI) additives? A comparison table in terms of price, toxicity, viscosity, dielectric constant and other parameters is needed to highlight the advantages of this electrolyte additives.

Our response: Thanks a lot for the reviewer's insightful suggestion. 1,3-Dimethyl-2-imidazolidinone (DMI) is a commonly used highly polar, aprotic solvent in industry, capable of miscibility with water in any proportion. It possesses characteristics such as high boiling point, low toxicity, low melting point, acid and alkali resistance, and excellent solubility for both inorganic and organic substances. A comparison of its basic properties with currently reported Zn anode electrolyte additives is presented in Table 1. As indicated in the **Table. R1**, in addition to the aforementioned advantages, when

DMI is employed as an additive in Zn anode electrolytes, only a very small quantity is required to achieve remarkably superior electrochemical performance compared to other reported additives. This contributes to maintaining the desirable properties of aqueous Zn cathode electrolytes, including non-flammability, low viscosity (1.4 mPa·s), high conductivity (43.1 mS cm⁻¹), and low electrolyte cost. Therefore, based on these findings, we consider DMI to be a highly promising additive for aqueous Zn ion battery electrolytes.

Table R1. Comparison of properties of electrolyte additives for Zn anodes.

Additives	Price(\$/ton)	Toxicity	Viscosity (mPa · s)	Dielectric constant	Concentration	Reference
DMI	1106.9	Low toxicity	1.944	37.6	0.01 M L ⁻¹	This work
THF	2739.8	Oncogenic, flammable	0.58	7.6	25%	Adv. Mater. 35 , e2210051 (2023).
NMP	3103.0	Reproductive toxicity, flammable	1.65	32	5%vol	Angew. Chem. Int. Ed. Engl. 61 , e202212839 (2022)
DOTf	27641800	Irritant	N/A (solid)	N/A (solid)	50	Joule 5 , 1119-1142 (2021)
Me₃EtNOTF	N/A (unsold)	N/A	N/A (solid)	N/A (solid)	0.5 m kg ⁻¹	Nat. Nanotechnol. 16 , 902–910 (2021)
DMF	1193.4	Low toxic, irritant, flammable	0.92	36.7	33%vol	ACS Energy Lett. 8 , 1613-1625 (2023).
LiBOB	262564.7	Toxic, irritant	N/A (solid)	N/A (solid)	0.1 M L ⁻¹	Angew. Chem. Int. Ed. 62 , e202311032 (2023)
IU	31826.0	Irritant	N/A (solid)	N/A (solid)	0.5 m kg ⁻¹	DOI: 10.1039/D3EE01580G

TMACI	4146.3	Irritant	N/A (solid)	N/A (solid)	10 m kg ⁻¹	Nat Sustain 6 , 806–815 (2023)
HTFSI	1214.95	Toxic, Corrosive	N/A (solid)	N/A (solid)	0.1 m kg ⁻¹	Nat Commun 15 , 4303 (2024).

Comment 2: Is it possible to regulate the thickness or composition of this bilayer SEI?

This may further improve the final electrochemical performances.

Our response: Thanks for the reviewer's constructive comments. Since the formation of the double-layer SEI is due to the electrochemical reduction of DMI, we believe that by changing the concentration of DMI added to the electrolyte, we can control the thickness of the SEI. For example, at low concentrations of DMI, the SEI thickness might be thinner, whereas at high concentrations, the SEI thickness might be thicker.

Regarding the composition regulation of the SEI on the zinc anode, this may be related to the type of electrolyte additives used. Different types of additives can form SEI layers with different chemical compositions and physical properties. Based on currently reported results (**Fig. R1**), it is generally believed that F-containing additives may lead to a ZnF₂-rich SEI (*Nat. Nanotechnol.* **16**, 902–910 (2021)), while Cl-containing additives may result in a Cl-rich SEI (*Nat Sustain* **6**, 806–815 (2023)). However, there is no established rule on how to design and regulate specific compositions and structures of SEI layers according to one's requirements.

Fig. R1. Research on the formation of SEIs with different compositions. **a**, ZnF₂-rich SEI (*Nat. Nanotechnol.* **16**, 902–910 (2021)). **b**, Cl-rich SEI (*Nat Sustain* **6**, 806–815 (2023)).

We also hope that future research will focus more on this area to establish general principles for designing SEI layers on Zn anodes. This is a very interesting and meaningful topic, and it will be part of our future research direction.

Comment 3: According to the proposed bilayer-SEI formation mechanism, the H^+ ions will be consumed and the pH of electrolyte will change. This will promote the zinc dendrites growth. So, the pH variations during electrochemical process will be very important. Could do please provide related data and give corresponding explanation?

Our response: Thanks a lot for the reviewer's insightful suggestion. According to previous literature reports, the formation of SEI is generally considered to occur primarily during the first cycle of metal deposition. Based on our proposed mechanism for SEI formation, we hypothesize that a significant amount of H^+ ions is required to participate in the reaction at the anode side during the SEI formation process. Therefore, a H^+ -rich region forms near the anode, leading to a decrease in pH, especially during the reduction of SO_4^{2-} to S^{2-} . To investigate this, we designed an in-situ pH detection setup to monitor and analyze the pH changes of the electrolyte during the initial zinc deposition process. The in-situ pH detection setup is shown in **Fig. R3a**, with the pH monitoring probe placed close to the anode.

As shown in **Fig. R3b**, the initial pH of the 2 M $ZnSO_4$ + 10 mM DMI electrolyte is 3.79. When the current is applied and Zn metal begins to deposit on the anode, the pH of the electrolyte near the anode starts to drop. Throughout the deposition process, the pH gradually decreases to 3.7, indicating the formation of an H^+ -rich region near the anode during Zn deposition. This strongly supports the validity of our proposed SEI formation mechanism. When the deposition ends, the pH of the electrolyte stabilizes and returns to 3.75, which is slightly lower than the initial pH. The reason for the slow and small pH change is likely due to the low concentration of DMI (only 10 mM) added, resulting in minimal H^+ consumption.

To better illustrate the effect of pH on SEI formation, we further tested the pH changes of the electrolyte near the anode in 2 M $ZnSO_4$ electrolyte, where SEI does not form. The initial pH of the 2M $ZnSO_4$ electrolyte is 4.07. In stark contrast, during Zn

deposition, the electrolyte pH rapidly increases to 4.96. This may be due to the vigorous HER (hydrogen evolution reaction) occurring during zinc deposition in ZnSO_4 electrolyte, where a large amount of H^+ is quickly consumed, causing the electrolyte pH to rise sharply. When the deposition ends, the pH of the electrolyte returns to 4.11, which is slightly higher than the initial pH.

Fig. R3. The results of in-situ pH test during Zn deposition for the first cycle. **a**, The digital photo of in-situ pH detection device. **b**, Representative trend profiles of pH changes during Zn deposition in 2 M ZnSO_4 +10 mM DMI electrolyte. **c**, Representative trend profiles of pH changes during Zn deposition in 2 M ZnSO_4 electrolyte.

Our revision: According to the reviewer's comment, we have supplemented with in-situ pH testing to characterize the pH changes on the Zn anode side as Supplementary

Fig. 21, further verifying the mechanism of SEI formation. The details as follows: “*In the meanwhile, a small portion of SO_4^{2-} adsorbed on the surface of the Zn anode undergoes reduction to S^{2-} in the electron-driven and proton-driven processes^{34, 35}. The S^{2-} further combines with Zn^{2+} to form ZnS ³⁶. In-situ pH monitoring showed a decrease in pH at the Zn anode during the Zn deposition process, forming a locally acidic environment, which further corroborates the formation of ZnS (Supplementary Fig. 21 and Supplementary Video 1).*”. Please see the highlighted part in yellow background on page 12 in the revised manuscript and Supplementary Fig. 21 on page 22 in the revised Supplementary Information.

Comment 4: The authors states that “The bilayer SEI enables dense Zn deposition (actual thickness: 18.4 μm) close to the theoretical Zn deposition thickness of 17.1 μm .” Could do please give some explanation why the actual thickness of zinc deposition is larger than theoretical one?

Our response: We appreciate the reviewer's good suggestion. We would like to point out that we calculated the theoretical deposition thickness based on the theoretical volumetric capacity of Zn and the areal capacity of Zn deposition we set in the experiment. In Fig. 5g-5j, we set the areal capacity of Zn deposition on the Cu foil substrate as 10 mAh cm^{-2} . Given that the theoretical volumetric capacity of Zn is 5855 mAh cm^{-3} , thus, the theoretical Zn deposition thickness at 10 mAh cm^{-2} is calculated as $10/5855*10000=17.1 \mu\text{m}$. However, the actual thickness of Zn deposition based on the bilayer SEI is about 18.4 μm , slightly higher than that of theoretical value. The theoretical deposition thickness of Zn is calculated based on ideal conditions, such as uniform Zn^{2+} concentration on the electrode surface, uniform Zn deposition rate, and absence of side reactions. However, practical electrodeposition processes often cannot meet these ideal conditions. For example, changes in Zn^{2+} concentration on the electrode surface due to Zn^{2+} consumption during deposition can lead to variations in Zn deposition rates at different locations, thereby affecting the final deposition thickness (*PNAS* **120**, e2307847120 (2023)). Additionally, hydrogen evolution reaction (HER) can also contribute to deposition thickness exceeding theoretical values. While

our designed bilayer SEI can suppress HER occurrence, it cannot completely eliminate it. Due to the influence of Zn deposition potential, HER inevitably accompanies zinc electrodeposition. Residual hydrogen on the electrode surface may increase the electrode surface volume, resulting in the actual thickness exceeding theoretical thickness (*Energy Environ. Sci.* **17**, 1975-1983 (2024)).

Comment 5: In the 359th line, you don't need to define the ZUR for the second time. Moreover, the record zinc utilization of zinc anode with bilayer SEI is one of highlights in this work. The detailed test and calculation methods should be given in the Methods part. The thickness and size of used zinc foil should be involved in Methods part.

Our response: Thanks for the reviewer's good comment and kind suggestion. We agree with the reviewer's suggestion regarding the unnecessary redefinition of "ZUR," and have accordingly made the corresponding modifications in the revised manuscript.

In this work, the ZUR is defined as the design cycling capacity divided by the actual capacity of the Zn anode used, i.e. $ZUR = \text{set capacity of the experiment} / \text{actual capacity of the Zn foil used}$. The specific details of the experiment are as follows: prior to assembling Zn-Zn symmetric cells with high ZUR, the Zn foil was first cleaned with deionized water and anhydrous ethanol. The surface was then gently polished with fine sandpaper to remove the oxide layer and impurities, followed by another cleaning with water and anhydrous ethanol. The treated Zn foil was cut into 12 mm diameter discs and weighed. The actual capacity of the Zn anode used was calculated based on its weight and the theoretical specific capacity of Zn. Assuming the weight of the Zn anode is x g and the cycling capacity is y mAh, the ZUR can be calculated using the formula: $ZUR = y / (x * 820)$.

Our revision: According to the reviewer's comment, we have added calculation methods of Zn anode with high ZUR in the revised manuscript. The detailed revision is shown as follows: "*Prior to assembling Zn-Zn symmetric cells with high ZUR, the Zn foil was first cleaned with deionized water and anhydrous ethanol. The surface was then gently polished with fine sandpaper to remove the oxide layer and impurities,*

*followed by another cleaning with water and anhydrous ethanol. The treated Zn foil was cut into 12 mm diameter discs and weighed. The actual capacity of the Zn anode used was calculated based on its weight and the theoretical specific capacity of Zn (820 mAh g⁻¹). At this point, the ZUR is defined as the design cycling capacity divided by the actual capacity of the Zn anode used. Assuming the weight of the Zn anode is x g and the cycling capacity is y mAh, the ZUR can be calculated using the formula: $ZUR=y/(x*820)$.* Please see the highlighted part in yellow background on page 22, page 23 in the revised Supplementary Information.

REVIEWERS' COMMENTS

Reviewer #1 (Remarks to the Author):

Recommendation: Publish as is; no revisions needed.

Reviewer #2 (Remarks to the Author):

The authors have addressed the concerns very well. However, it is still lack of novelty for Nature Communications. Although the author provided a detailed analysis and characterization on SEI, the initial idea was the main additive, which have already been reported. Therefore, the highlights of this article are not sufficient to meet the high requirements of NC.

Reviewer #3 (Remarks to the Author):

The authors already addressed my concerns by additional experiments. and I have no further comments.